# Efficient Ensemble Conditional Independence Test Framework for Causal Discovery

**Zhengkang Guan, Kun Kuang**[*]
College of Computer Science and Technology, Zhejiang University
`zhengkang.guan@zju.edu.cn, kunkuang@zju.edu.cn`

## Abstract

Constraint-based causal discovery relies on numerous conditional independence tests (CITs), but its practical applicability is severely constrained by the prohibitive computational cost, especially as CITs themselves have high time complexity with respect to the sample size. To address this key bottleneck, we introduce the Ensemble Conditional Independence Test (E-CIT), a general-purpose and plug-and-play framework. E-CIT operates on an intuitive divide-and-aggregate strategy: it partitions the data into subsets, applies a given base CIT independently to each subset, and aggregates the resulting p-values using a novel method grounded in the properties of stable distributions. This framework reduces the computational complexity of a base CIT to linear in the sample size when the subset size is fixed. Moreover, our tailored p-value combination method offers theoretical consistency guarantees under mild conditions on the subtests. Experimental results demonstrate that E-CIT not only significantly reduces the computational burden of CITs and causal discovery but also achieves competitive performance. Notably, it exhibits an improvement in complex testing scenarios, particularly on real-world datasets.

## 1 Introduction

Conditional independence testing (CIT) serves as a foundational tool in statistics and machine learning, particularly central to causal discovery algorithms (Spirtes et al., 2000; 1995; Glymour et al., 2019; Vowels et al., 2022), which fundamentally rely on CIT to examine whether variables $X$ and $Y$ are independent given a conditioning set $Z$. Formally, it evaluates the following hypotheses:

$$H_0 : X \perp\!\!\!\perp Y \mid Z \quad \text{versus} \quad H_1 : X \not\perp\!\!\!\perp Y \mid Z.$$

However, the heavy reliance of constraint-based causal discovery on numerous CITs creates a severe computational bottleneck, significantly limiting its practical use. While many studies (Akbari et al., 2021; Mokhtarian et al., 2021; 2023; 2025; Shiragur et al., 2024; Rohekar et al., 2021) have focused on reducing the number of CITs to streamline the discovery process, a more fundamental challenge lies in the high time complexity of CITs themselves (Zhang et al., 2011; Scetbon et al., 2022). Despite some research (Strobl et al., 2019; Schacht & Huang, 2025) on mitigating the cubic time complexity of the popular kernel-based conditional independence test (KCIT) (Zhang et al., 2011), Shah & Peters (2018) demonstrate that no single CIT is uniformly effective across all conditional dependence structures. Thus, a critical open question is how to generally reduce the computational cost of CITs while preserving their testing power.

To address this challenge, we propose the Ensemble Conditional Independence Test (E-CIT), a general-purpose plug-and-play framework that can be seamlessly applied to existing CIT methods to mitigate the computational burden while maintaining competitive performance. E-CIT adopts an intuitive divide-and-aggregate strategy: given a CIT method, it partitions the dataset into multiple subsets, conducts independent tests on each subset, and aggregates the resulting p-values. For this combination, we introduce a novel method based on the properties of stable distributions, which is theoretically consistent under mild conditions on the subtests, and ensures the reliability of the overall procedure. When the subset size is fixed, this strategy controls the computational complexity of the base CIT to linear in the sample size.

---

[*]Corresponding author.

The main contributions of this paper are summarized as follows:

- We introduce E-CIT, a general-purpose divide-and-aggregate framework that systematically mitigates the computational complexity of CITs, thereby addressing a fundamental computational bottleneck in causal discovery with respect to sample size.

- We develop a novel p-value combination method grounded in the closure property of stable distributions, which offers validity and consistency under mild conditions on the subtests, while remaining flexible across different settings.

- Through extensive experiments on both synthetic and real-world datasets, we show that E-CIT yields substantial efficiency gains while achieving competitive performance, especially in challenging heavy-tailed or real-world scenarios.

## 2 RELATED WORK

### 2.1 CONDITIONAL INDEPENDENCE TESTING

We briefly review several representative and recent approaches to CIT, while referring to Li & Fan (2020) for a comprehensive overview. A typical approach in CIT is to define criteria for conditional independence. One of the most widely used measures is Conditional Mutual Information (CMI) (Runge, 2018; Mukherjee et al., 2020; Jamshidi et al., 2024), along with several other metrics (Yu et al., 2020; Wang et al., 2015; Cai et al., 2022). Sen et al. (2017) reformulate CIT as a binary classification problem and apply modern classifiers for hypothesis testing. Recently, the Conditional Randomization Test (CRT) (Candes et al., 2018) has inspired several new methods. For example, Bellot & van der Schaar (2019); Shi et al. (2021) utilize GANs for conditional sampling, whereas Li et al. (2023a;b) employ nearest-neighbor sampling techniques. These methods are particularly effective for handling large conditioning sets.

KCIT (Zhang et al., 2011) is a widely used CIT method leveraging reproducing kernel Hilbert spaces (RKHS), and has inspired many kernel-based extensions (Doran et al., 2014; Scetbon et al., 2022; Zhang et al., 2022; Pogodin et al., 2024). Efforts to accelerate CITs have primarily focused on approximations of KCIT, notably RCIT (Strobl et al., 2019) and FastKCIT (Schacht & Huang, 2025). RCIT employs random Fourier features for efficient approximation, while FastKCIT partitions the dataset with Gaussian mixture models of the conditioning variable $Z$. Although similar in spirit, FastKCIT is specifically tailored to KCIT rather than serving as a general framework like E-CIT.

### 2.2 COMBINATION TEST

The problem of combining individual p-values into an overall test has long been central in statistics, with important applications in fields such as genomics. Consider the scenario where the same hypothesis is tested $m$ times, yielding $m$ corresponding p-values $p_1, \ldots, p_m$. Classical methods for combining these p-values are summarized in Table 1, where $\Phi(\cdot)$ denotes the cumulative distribution function of the standard normal distribution. Heard & Rubin-Delanchy (2018) investigate the conditions under which each classical method for combining p-values is most appropriate.

Table 1: Classical p-value combination methods

| Method | Formula |
| --- | --- |
| Tippett (1931) | $\min(p_i)$ |
| Edgington (1972) | $\sum_{i=1}^{m} p_i$ |
| Fisher (1934) | $-2\sum_{i=1}^{m} \ln p_i$ |
| Pearson (1933) | $-2\sum_{i=1}^{m} \ln(1 - p_i)$ |
| Mudholkar & George (1979) | $\sum_{i=1}^{m} \ln[p_i/(1 - p_i)]$ |
| Stouffer et al. (1949) | $\sum_{i=1}^{m} \Phi^{-1}(p_i)$ |
| Lipták (1958) | $\sum_{i=1}^{m} \Phi^{-1}(1 - p_i)$ |

Recently, the problem of combining multiple p-values has received renewed attention (Vovk & Wang, 2020; Geistkemper, 2024), particularly in high-dimensional settings with dependent tests, as is common in biostatistics. Liu & Xie (2020) propose a Cauchy-based method using inverse probability weighting, which performs well under such conditions (Long et al., 2023). Building on this technique, Liu et al. (2024) develop an ensemble testing method, and Ling & Rho (2022) further generalize this approach using stable distributions. Notably, these specific methods are explicitly designed for traditional parametric settings, primarily for whole-genome sequencing (WGS) association studies.

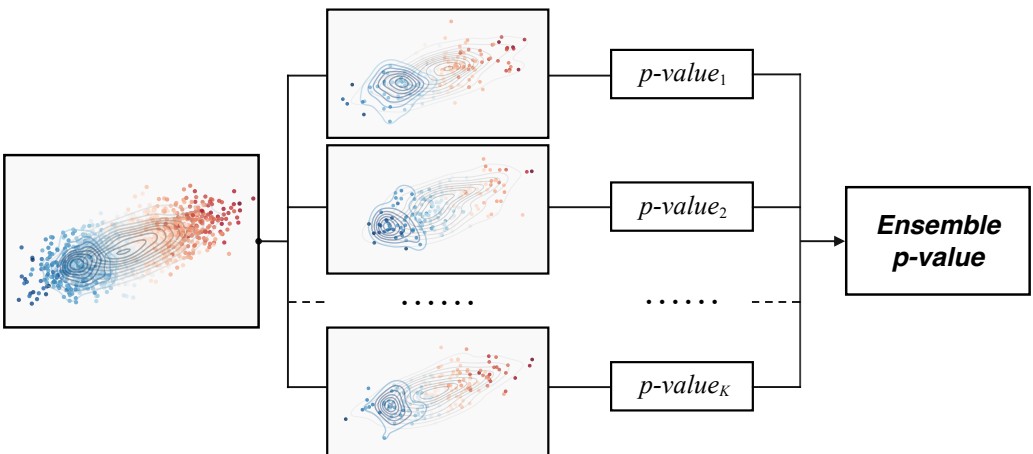

Figure 1: Overview of the E-CIT framework. Each scatter plot displays samples of variables $X$ and $Y$, with color indicating the value of $Z$. Despite smaller subset sizes, the marginal dependence (black contours) and conditional independence given $Z$ (blue contours) remain clearly distinguishable.

## 3 METHOD

### 3.1 ENSEMBLE CONDITIONAL INDEPENDENCE TEST FRAMEWORK

Given $n$ samples from a joint distribution over $X$, $Y$, and $Z$, our goal is to test whether $X \perp\!\!\!\perp Y \mid Z$ using an arbitrary base CIT method. However, as the sample size $n$ grows, the computational cost of many CIT methods can become prohibitive due to their high complexity. Inspired by ensemble learning, we propose the Ensemble Conditional Independence Test (E-CIT) framework to address this issue, as shown in Figure 1. We partition the entire dataset into $K$ subsets of size $n_k$, where $n = Kn_k$. The base CIT is applied independently to each subset, yielding p-values $\{p_1, \ldots, p_K\}$, which are then combined into a final p-value. When $n_k$ is fixed, this ensures that the overall computational cost scales linearly with $n$, regardless of the original complexity of the CIT method.

As illustrated in Figure 1, subsets of sufficient size can effectively capture conditional dependence. However, unlike classical parametric hypothesis tests and their associated p-value combination methods (Liu & Xie, 2020; Ling & Rho, 2022), CITs have a more complex alternative hypothesis, leading to p-value distributions that vary significantly across data-generating mechanisms and base CIT methods. This variability poses a challenge for p-value combination, since the statistical properties of a combination method depend heavily on the alternative distribution of p-values (Heard & Rubin-Delanchy, 2018). Therefore, to ensure broad applicability across scenarios and methods, a flexible aggregation strategy that maintains statistical properties under diverse conditions is essential. In the next section, we propose such a method based on the properties of stable distributions for the E-CIT framework.

### 3.2 COMBINING P-VALUES VIA STABLE DISTRIBUTIONS

We utilize the properties of stable distributions to combine p-values to construct the aggregation strategy. We begin with a brief introduction to stable distributions, as detailed in Nolan (2012; 2020).

**Definition 1** (Stable Distribution). *A random variable $X$ follows a Stable Distribution with parameters $\alpha \in (0, 2]$, $\beta \in [-1, 1]$, $\gamma > 0$, and $\delta \in \mathbb{R}$, denoted as $X \sim \mathbf{S}(\alpha, \beta, \gamma, \delta)$, if its characteristic function is given by:*

$$\mathbb{E}\left[\exp(iuX)\right] = \begin{cases} \exp\left(-\gamma^\alpha |u|^\alpha \left[1 - i\beta \left(\tan \frac{\pi\alpha}{2}\right)(\text{sign}(u))\right] + i\delta u\right), & \alpha \neq 1 \\ \exp\left(-\gamma|u| \left[1 + i\beta \frac{2}{\pi}(\text{sign}(u)) \log |u|\right] + i\delta u\right), & \alpha = 1. \end{cases}$$

The formulations above represent one of the parameterizations of the characteristic function used to define stable distributions. The parameter $\alpha$, known as the stability parameter, controls the tail

heaviness, with smaller values of $\alpha$ corresponding to heavier tails. The parameter $\beta$ is the skewness parameter, where $\beta = 0$ corresponds to a symmetric distribution. In this case, the normal and Cauchy distributions arise when $\alpha = 2$ and $\alpha = 1$, respectively, whereas the skewed Lévy distribution corresponds to $\beta = 1$ and $\alpha = 0.5$. The scale parameter $\gamma$ determines the spread of the distribution, and the location parameter $\delta$ shifts the distribution along the real axis.

The most important property of stable distributions is their generalization of closure under summation, as the sum of independent stable-distributed random variables remains stable (detailed in Appendix A). In our method, we utilize the closure property in a specific case, as shown below.

**Proposition 1.** *Let* $X_1, X_2, \ldots, X_J$ *be independent and identically distributed (i.i.d.) random variables following a stable distribution:*

$$X_j \sim \mathbf{S}(\alpha, \beta, \gamma, \delta), \quad j = 1, 2, \ldots, J.$$

*Then, the normalized sum*

$$S_J = \frac{X_1 + \cdots + X_J}{J}$$

*also follows a stable distribution:* $S_J \sim \mathbf{S}\left(\alpha, \beta, \gamma', \delta\right)$*, where* $\gamma' = J^{\frac{1}{\alpha} - 1} \gamma$.

This elegant property gives rise to the name of the stable distribution. Building upon this property, we define the core of our proposed method:

**Definition 2** (Ensemble Test)**.** *Given a set of i.i.d. p-values* $p_1, p_2, \ldots, p_K$ *derived from independent and identical subtests* $\mathcal{H}$*, the ensemble test* $\mathcal{H}_e\left(\mathcal{H}, K; \alpha, \beta, \gamma, \delta\right)$ *is defined by the test statistic* $T_e$*:*

$$T_e = \frac{1}{K} \sum_{k=1}^{K} F_S^{-1}(p_k),$$

*where* $F_S^{-1}$ *is the inverse cumulative distribution function (CDF) of the stable distribution* $\mathbf{S}(\alpha, \beta, \gamma, \delta)$*. It is evident that we obtain the lower-tail p-value, referred to as the ensemble p-value* $p_e$*, given by:*

$$p_e = F_{S'}(T_e),$$

*where* $F_{S'}$ *is the CDF of the stable distribution* $\mathbf{S}(\alpha, \beta, \gamma', \delta)$ *with* $\gamma' = K^{\frac{1}{\alpha} - 1} \gamma$.

The ensemble test combines individual p-values into a single test statistic by leveraging the properties shown in Proposition 1. This approach offers greater flexibility by allowing adaptive selection of the stable distribution parameters $\alpha, \beta, \gamma, \delta$ to accommodate different types of CIT and underlying conditional dependence structures in the data. Among these parameters, $\alpha$ controls the tail heaviness of the stable distribution and has the greatest influence on its CDF $F_S$. Therefore, in practice, we recommend fixing $\beta, \gamma, \delta$ and varying only $\alpha$, which provides a simple yet effective way to adjust the flexibility of E-CIT.

It is important to distinguish our approach from related work such as Ling & Rho (2022). Structurally, our method is a generalized form of Stouffer et al. (1949), whereas Ling & Rho (2022) generalizes Liu & Xie (2020) and Lipták (1958). More importantly, our subsequent theoretical analysis is tailored for the challenges of CITs and, consequently, makes no parametric assumptions on the form of the subtests (such as the normality of subtests' statistics (Liu & Xie, 2020; Ling & Rho, 2022)).

To formally establish the reliability of our method, we present its key theoretical properties below (see Appendix B for detailed proofs).

**Theorem 1.** *The ensemble test* $\mathcal{H}_e$ *(for exact subtest p-values) satisfies the following properties:*

1. ***Validity****: Under the null hypothesis, the ensemble p-value is uniformly distributed on* $[0, 1]$*, ensuring Type I error control.*

2. ***Admissibility****: The ensemble test is admissible, indicating that no other test uniformly outperforms it in terms of error rates and decision-making optimality.*

3. ***Unbiasedness***[1]*: The ensemble test is unbiased if its subtests are unbiased, meaning the ensemble does not compromise the unbiasedness of the individual subtests.*

---

[1]Unbiasedness in hypothesis testing is defined as the rejection probability under the alternative hypothesis being at least the pre-specified significance level (Lehmann & Romano, 2005), distinct from unbiasedness in estimation.

Theorem 1 ensures that our ensemble test is valid for exact base p-values. However, we acknowledge that due to the challenges inherent to CITs, these p-values are often approximate. This implies that the guarantees of Theorem 1 may not hold exactly in practice, a point we discuss further in Appendix F.

We further examine the ensemble test's power. Let $\alpha_e$, $\beta_e$, and $\pi_e = 1 - \beta_e$ denote the Type I error, Type II error, and power of the ensemble test, respectively. The following theorem establishes a key result that describes the power of our ensemble test.

**Lemma 1.** *Assume that $F_S^{-1}(p_k^{H_1})$ is integrable. The power of the ensemble test $\mathcal{H}_e\,(\mathcal{H}, K; \alpha, \beta, \gamma, \delta)$ approaches 1 as $K \to \infty$, i.e., $\lim_{K\to\infty} \pi_e = 1$, if the following condition holds:*

$$\mathbb{E}\left[F_S^{-1}(p_k^{H_1})\right] < F_{S'}^{-1}(\alpha_e),$$

*where $p_k^{H_1}$ for $k = 1, 2, \ldots, K$ are i.i.d. p-values from the subtest $\mathcal{H}$ under the alternative hypothesis.*

Lemma 1 establishes a sufficient condition for the power of our ensemble test to converge to 1. However, the conditions stated are not directly interpretable and may be difficult to verify. Thus, we present the following theorem, which provides more relaxed and practically verifiable conditions for convergence.

**Theorem 2.** *Consider the ensemble test $\mathcal{H}_e\,(\mathcal{H}, K; \alpha, \beta, \gamma, \delta)$, and assume that $F_S^{-1}(p_k^{H_1})$ is integrable. If the following conditions hold:*

1. *$\mathbb{E}[p_k^{H_1}] \leq \alpha_e$,*

2. *$f_1(p) \geq f_1(1 - p)$ for $p \in \left[0, \frac{1}{2}\right]$, where $f_1$ is the probability density function of $p_k^{H_1}$,*

3. *$\alpha \geq 1, \beta = \delta = 0$.*

*Then, we have $\lim_{K\to\infty} \pi_e = 1$.*

**Remark 1.** Theorem 2 highlights the reliability of the E-CIT framework. It shows that E-CIT not only preserves the consistency of the base CITs but also offers a potential way to improve the power of methods lacking theoretical consistency guarantees. More importantly, while many existing CITs have consistency guarantees (Scetbon et al., 2022; Jamshidi et al., 2024), their underlying assumptions can be difficult to satisfy in complex scenarios (Appendix C). In contrast, the consistency of E-CIT established by Theorem 2 does not directly impose assumptions on the testing scenario itself. Instead, it only requires the individual subtests to be reasonably effective. This property enhances the general applicability of E-CIT in challenging situations (see experimental results in Section 4).

**Remark 2.** The three conditions in Theorem 2 can be easily satisfied in practice. The first condition requires the performance of individual subtests, specifically that the expected p-value of the subtest $\mathcal{H}$ under the alternative hypothesis $H_1$ is below the significance level. This condition is mild and can be directly translated to requiring that the power of $\mathcal{H}$ exceeds a threshold determined by $f_1$, which can be as low as 0.5 in edge cases. See further illustrations in Appendix D.

The second condition concerns the shape of the p-value distribution under $H_1$, requiring the density on the left side of $f_1$ to exceed the symmetric value on the right. This is natural since p-values under $H_1$ tend to concentrate near 0, and it is automatically satisfied when the first condition holds and p-values are approximated by a Beta distribution (Heard & Rubin-Delanchy, 2018) (Appendix D).

The third condition restricts the stable distribution used in $\mathcal{H}_e$. The requirement $\alpha \geq 1$ ensures the tail is no heavier than Cauchy distribution, which aligns with statistical intuition about tail behavior (Liu & Xie, 2020). Although $\beta = \delta = 0$ can be relaxed in theory (see Eq.(3), Appendix B.3), we fix them to simplify both the proofs and implementation, while using $\alpha$ to control tail heaviness in practice.

**Remark 3.** It is important to note that this desirable convergence property occurs with respect to the number of subtests $K$, rather than the sample size $n$. Moreover, the condition ensuring convergence primarily imposes requirements on the effectiveness of the individual subtests. Therefore, for a fixed total sample size, the ensemble approach benefits from increasing $K$ only if the effectiveness of each subtest can be maintained. A simple increase in $K$ alone may not improve performance.

Lastly, we discuss the rationale for maintaining flexibility in our E-CIT framework. Since valid p-values follow a uniform distribution under the null hypothesis, the Neyman-Pearson lemma dictates that the uniformly most powerful test statistic for combining p-values should correspond to a

monotonic transformation of $-\sum_{k=1}^{K} \log f_1(p_k)$ (Casella & Berger, 2024; Heard & Rubin-Delanchy, 2018). Therefore, the ensemble test is optimal when

$$\sum_{k=1}^{K} F_S^{-1}(p_k) = g\left(-\sum_{k=1}^{K} \log f_1(p_k)\right),$$

where $g$ is an arbitrary monotonic function. However, for CIT, unlike traditional parametric tests, the conditional dependence structure of the data under the alternative hypothesis $H_1$ can lead to different distributions of $p_k^{H_1}$ even with the same CIT method. Therefore, our E-CIT allows the flexibility of adjusting $\alpha$ to concisely control $F_S$, enabling the test statistic to satisfy the above condition as closely as possible. However, we acknowledge that the theoretically optimal choice of $\alpha$ is context-dependent and requires further analysis for specific CIT methods (see Appendix F for a discussion).

## 4 EXPERIMENTS

In this section, we comprehensively evaluate the effectiveness of E-CIT. We first demonstrate its ability to reduce computational costs while maintaining performance (Section 4.1), followed by its broad applicability across different CIT methods (Section 4.2) and strong performance on real-world datasets (Section 4.3). Additionally, we apply E-CIT in causal discovery (Section 4.4). Other results, including the impact of subset size and the advantages of our p-value combination method for CIT, can be found in Appendices E.6 and E.7.

We conduct our synthetic experiments under the post-nonlinear model, following the setup of prior works (Zhang et al., 2011; Doran et al., 2014; Bellot & van der Schaar, 2019; Scetbon et al., 2022; Li et al., 2023a;b). Specifically, we consider the null hypothesis $H_0 : X \perp\!\!\!\perp Y \mid Z$ and the alternative hypothesis $H_1 : X \not\perp\!\!\!\perp Y \mid Z$, with data generated as follows:

$$H_0 : \quad X = f_X\left(W_X^\top Z + \varepsilon_X\right), \quad Y = f_Y\left(W_Y^\top Z + \varepsilon_Y\right)$$
$$H_1 : \quad X = f_X\left(W_X^\top Z + \varepsilon_X\right), \quad Y = f_Y\left(W_Y^\top Z + \beta_X X\right) + \varepsilon_Y$$

Here, $Z$ is drawn from either a standard normal or standard Laplace distribution. The weight matrices $W_X$ and $W_Y$ are initialized with entries from $U(0, 1)$ and column-normalized so each column sums to one, with $\beta_X$ set to 1. The nonlinear functions $f_X$ and $f_Y$ are randomly selected from $\{x, x^2, x^3, \tanh(x), \cos(x)\}$. The noise terms $\varepsilon_X$ and $\varepsilon_Y$ are i.i.d. samples from a standard Student's $t$, Laplace, or Cauchy distribution.

All CITs are evaluated at a significance level of 0.05. We compare each CIT with the original method and its ensemble version. For all ensemble tests, we fix the subtest sample size at $n_k = 400$, while the number of subtests $K$ varies with the total sample size, which ensures linear computational complexity. This value of $n_k$ is chosen based on empirical experience (Zhang et al., 2011; Scetbon et al., 2022; Runge, 2018), which indicates that CIT methods exhibit sufficient empirical behavior at this sample size, as further discussed in Appendix F. While $n_k = 400$ already yields good performance, our ablation study in Appendix E.6 indicates that further optimization is possible. Following Theorem 2, we set $\beta = \delta = 0$ and $\gamma = 1$. We use two values of $\alpha$ (1.75 and 2) to illustrate how the tail heaviness of the stable distribution affects performance, based on experiments reported in Appendix E.1.

### 4.1 EFFICIENCY THROUGH ENSEMBLE FRAMEWORK

In this experiment, we show that the ensemble framework reduces computational cost while maintaining competitive performance. We compare our ensemble-enhanced KCIT (E-KCIT) with RCIT (Strobl et al., 2019), FastKCIT (Schacht & Huang, 2025) (the only other methods known to accelerate CITs), and the original KCIT over 1000 independent trials. For all these methods, the kernel bandwidth is determined using the median heuristic. While advanced bandwidth optimization strategies represent an active area of research (Wang et al., 2025), we strictly adhere to the standard median heuristic as to ensure a controlled comparison and minimize potential confounding factors.

Figure 2 shows results under standard Student's $t$ (with two degrees of freedom), Cauchy, and Laplace noise, with $Z$ normally distributed, evaluating Type I error (left), test power (middle), and runtime (right). E-KCIT significantly reduces computational costs while maintaining competitive test power. Notably, under the more challenging heavy-tailed noise distributions (Figures 2a and 2b), E-KCIT

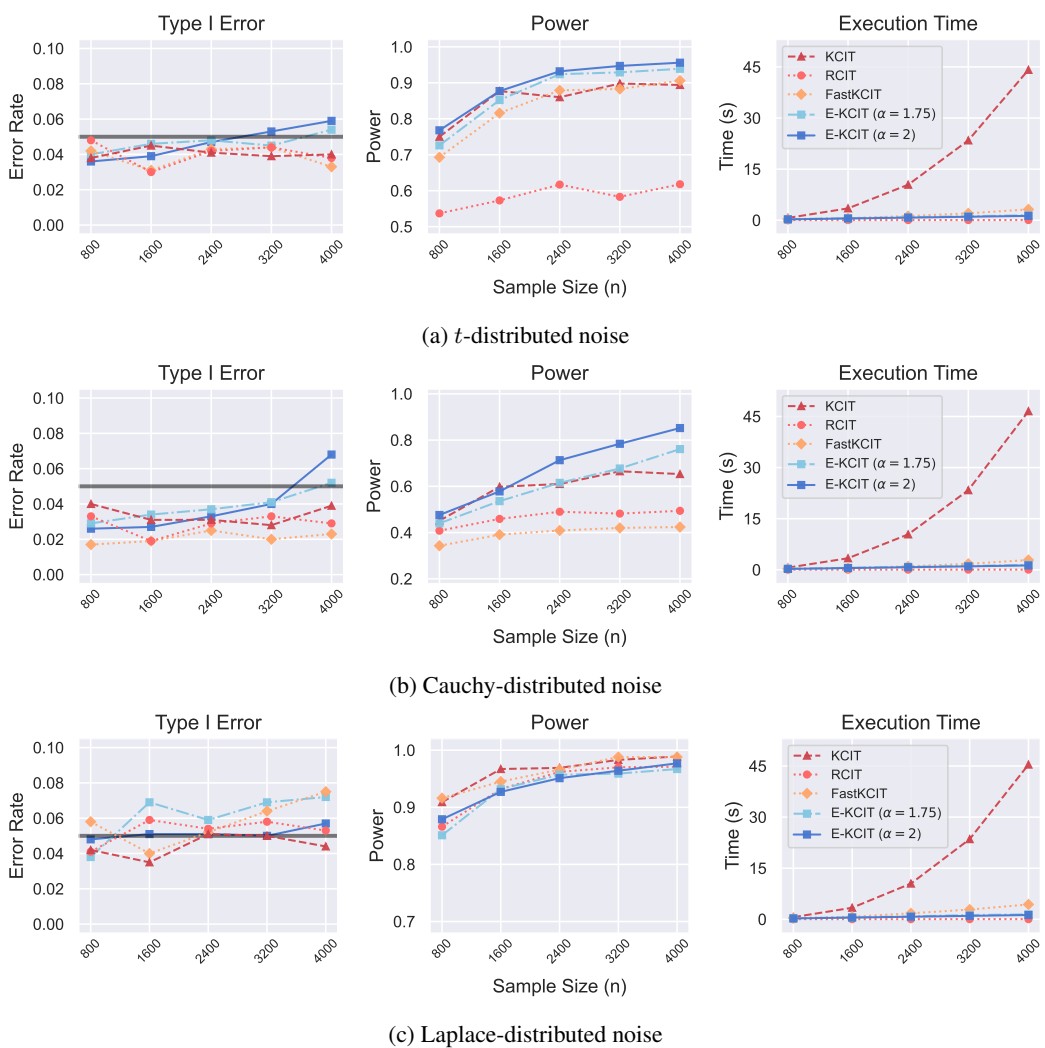

Figure 2: Comparison of Type I error (left; 0.05 significance level marked by solid black line), test power (middle), and runtime (right) for KCIT, RCIT, FastKCIT, and E-KCIT under different noise distributions.

demonstrates more consistent performance. Although all methods generally maintain the Type I error near the nominal significance level, the ensemble framework may slightly affect this control in some scenarios. As discussed in Appendix F, this effect is independent of the test's power. In some scenarios (as observed in subsequent experiments), it can lead to a more conservative Type I error.

## 4.2 ENSEMBLE EFFECTIVENESS ACROSS CONDITIONS

To evaluate the ensemble framework across diverse settings, we further compare five CIT methods: RCIT (Strobl et al., 2019), LPCIT (Scetbon et al., 2022), CMIknn (Runge, 2018), CCIT (Sen et al., 2017), and Fisher Z-test (FisherZ) (Fisher, 1921), in both their original (Orig.) and ensemble versions ($\alpha = 1.75$ and $\alpha = 2$). Simulations are conducted with sample sizes of 800, 1200, and 1600, with $Z$ sampled from a standard normal or Laplace distribution, and standard $t$-distributed noise with three different degrees of freedom (df=2, 3, 4).

Each setting is repeated 1000 times for RCIT, LPCIT, and Fisher Z-test, and 500 times for the more computationally intensive CMIknn and CCIT. Table 2, and additional results in Tables 4, 5, 6, 7 and 8 (Appendix E.2) report Type I error rates and test powers. Bold values indicate a statistically

Table 2: Results for $n = 1200$, Standard Normal $Z$. Bold (*Bold italics*) indicates the ensemble (original) version is statistically significantly better.

| df of $t$-distributed Noise | | df = 4 | | df = 3 | | df = 2 | |
|---|---|---|---|---|---|---|---|
| CIT Methods | | Type I | Power | Type I | Power | Type I | Power |
| RCIT | Orig. | 0.037 | 0.845 | 0.037 | 0.765 | 0.037 | 0.548 |
| | Ensemble $(\alpha=1.75)$ | 0.046 | **0.891** | 0.042 | **0.823** | 0.042 | **0.623** |
| | Ensemble $(\alpha=2.00)$ | 0.054 | **0.906** | 0.066 | **0.838** | 0.044 | **0.609** |
| LPCIT | Orig. | 0.054 | 0.742 | 0.054 | 0.659 | 0.021 | 0.422 |
| | Ensemble $(\alpha=1.75)$ | 0.042 | 0.755 | 0.031 | 0.675 | 0.013 | **0.447** |
| | Ensemble $(\alpha=2.00)$ | 0.054 | **0.767** | **0.026** | 0.669 | 0.017 | 0.418 |
| CMIknn | Orig. | 0.122 | 0.990 | 0.116 | 0.986 | 0.124 | 0.982 |
| | Ensemble $(\alpha=1.75)$ | 0.164 | 0.994 | 0.138 | 0.988 | 0.136 | 0.988 |
| | Ensemble $(\alpha=2.00)$ | 0.124 | 0.990 | 0.136 | 0.980 | 0.104 | 0.982 |
| CCIT | Orig. | 0.450 | *0.896* | 0.430 | *0.928* | 0.454 | *0.904* |
| | Ensemble $(\alpha=1.75)$ | **0.336** | 0.856 | **0.334** | 0.828 | **0.286** | 0.816 |
| | Ensemble $(\alpha=2.00)$ | **0.322** | 0.848 | **0.350** | 0.830 | **0.308** | 0.812 |
| FisherZ | Orig. | 0.217 | 0.695 | 0.144 | 0.613 | 0.093 | 0.510 |
| | Ensemble $(\alpha=1.75)$ | 0.197 | **0.766** | 0.138 | **0.659** | 0.094 | **0.561** |
| | Ensemble $(\alpha=2.00)$ | 0.213 | 0.719 | 0.124 | **0.656** | 0.078 | 0.508 |

Table 3: Performance comparison on the Flow-Cytometry dataset A (results for both $\alpha$ merged). Bold (*Bold italics*) indicates the ensemble (original) version is statistically significantly better.

| Method | | Precision | Recall | F1-score |
|---|---|---|---|---|
| KCIT | Orig. | 0.580 | 0.674 | 0.624 |
| | Ensemble | **0.730** | 0.664 | **0.695** |
| RCIT | Orig. | 0.684 | 0.647 | 0.665 |
| | Ensemble | **0.715** | **0.662** | **0.687** |
| LPCIT | Orig. | 0.740 | 0.649 | 0.691 |
| | Ensemble | **0.838** | **0.664** | **0.741** |
| CMIknn | Orig. | 0.880 | *0.698* | *0.779* |
| | Ensemble | 0.872 | 0.668 | 0.756 |
| CCIT | Orig. | 0.520 | *0.722* | 0.605 |
| | Ensemble | **0.618** | 0.680 | **0.646** |
| FisherZ | Orig. | 0.840 | 0.656 | 0.737 |
| | Ensemble | **0.852** | **0.699** | **0.767** |

significant improvement of the ensemble method over the original (based on a one-sided test at the 0.1 level), whereas bold italics denote cases where the original method performs better.

Across various simulation settings, the ensemble test consistently enhances the test power of RCIT, LPCIT, and the Fisher Z-test, while maintaining appropriate Type I error control. In contrast, the benefit for CMIknn is less pronounced. CCIT represents a special case: in our experiments, it fails to properly control the Type I error. Interestingly, applying the ensemble test significantly reduces the Type I error in this case, albeit with a minor reduction in power. We also observe that the choice of the E-CIT parameter $\alpha$ affects performance across different CIT methods and data configurations. For example, in Table 2 with $n = 1200$ and $Z$ drawn from a standard normal distribution, the ensemble framework with $\alpha = 2$ performs better for RCIT, whereas $\alpha = 1.75$ performs better for the Fisher Z-test. These observations align with our analysis in Section 3, highlighting the importance of flexibility in the E-CIT framework. We also evaluate the impact of the dimensionality of the conditioning set $Z$ in Appendix E.3.

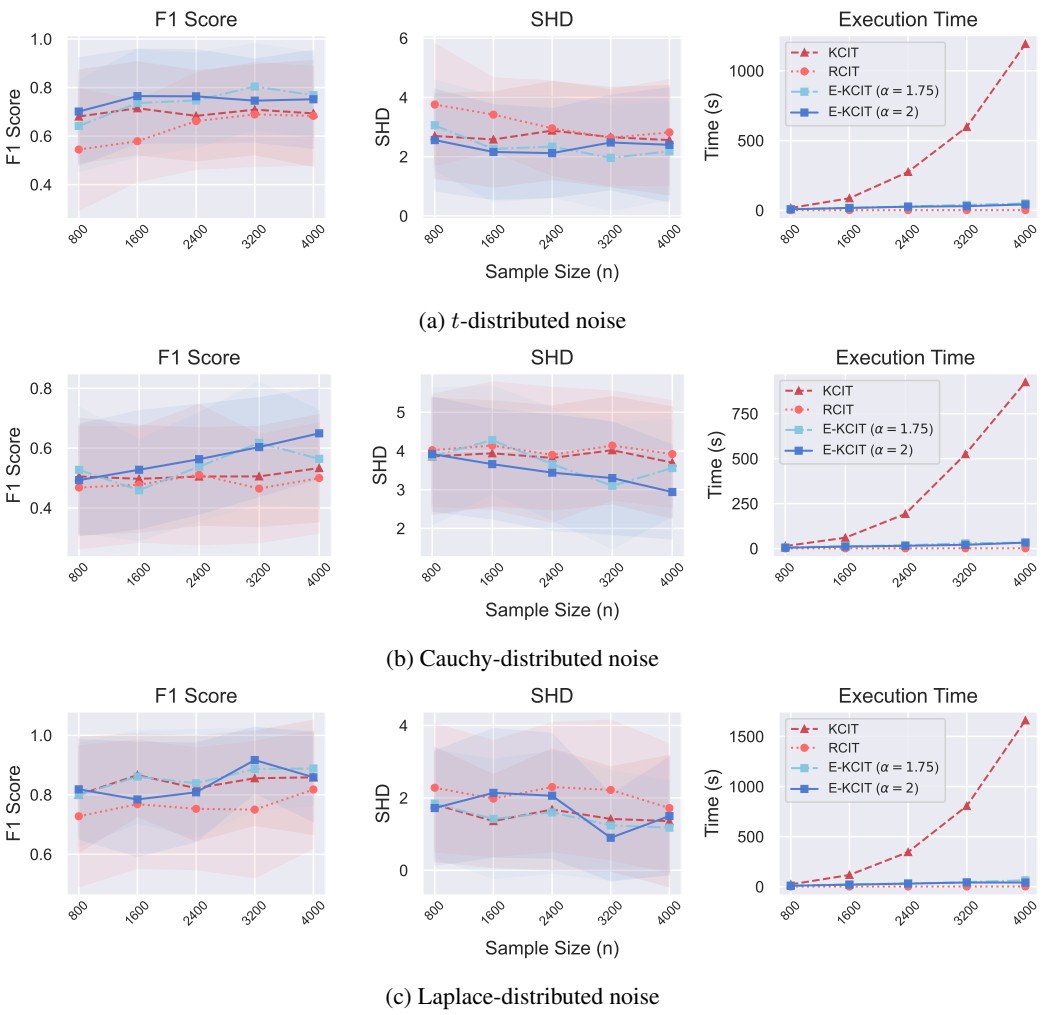

Figure 3: Comparison of causal discovery performance (F1-score, SHD, and runtime) of KCIT, RCIT, and E-KCIT under different noise distributions. Shaded areas indicate ±1 standard deviation.

### 4.3 REAL DATA EXPERIMENT: FLOW-CYTOMETRY DATASET

We evaluate E-CIT on the Flow-Cytometry dataset, with the experimental details provided in Appendix E.4. As shown in Table 3, the ensemble framework enhances the performance of most CIT methods on complex real-world datasets. While there is a slight performance drop for CMIknn, notable gains are observed for KCIT, RCIT, LPCIT, and FisherZ. For CCIT, consistent with Section 4.2, the ensemble framework improves its Type I error control, as reflected in higher precision. Overall, these results further confirm the broad applicability of E-CIT.

### 4.4 APPLICATION IN CAUSAL DISCOVERY

Similar to the experiments in Section 4.1, we evaluate the performance of E-KCIT against RCIT and KCIT on synthetic causal graphs generated with nonlinear functional mechanisms and additive noise. Detailed settings are provided in Appendix E.5. We consider Student's $t$ (df = 2), Cauchy, and Laplace noise distributions, with results shown in Figure 3. In most settings, E-KCIT outperforms RCIT and KCIT in both F1-score and Structural Hamming Distance (SHD), while its runtime remains comparable to RCIT. These results demonstrate that E-CIT is both practical and effective for causal discovery.

## 5 DISCUSSION

In this paper, we have introduced the Ensemble Conditional Independence Test (E-CIT), a general-purpose, plug-and-play framework that addresses the critical computational bottleneck in constraint-based causal discovery. By employing a divide-and-aggregate strategy, E-CIT can linearize the complexity of a base CIT. Moreover, based on stable distributions, our novel p-value combination method ensures statistical properties under mild conditions. Our theoretical and empirical findings suggest that the framework is especially effective in complex real-world scenarios. The significance of our work lies in its modularity: instead of proposing another specific CIT, we present a framework that enhances the scalability and can provide consistency for a broad class of CIT methods.

*Applicability and Scope.* While E-CIT provides a broadly applicable plug-and-play framework for reducing the computational burden of CITs, it is crucial to delineate its present theoretical and practical scope. First, our current analysis primarily assumes independent and identical subtests, yielding i.i.d. p-values. This condition may not be satisfied in certain scenarios (see the discussions on correlated p-values and distribution drifts in Appendix F). Second, although our p-value combination method ensures power consistency under mild conditions, its ultimate performance remains intrinsically tied to the base CIT. Consequently, while E-CIT can linearize complexity with respect to sample size, it does not resolve fundamental statistical challenges, such as the curse of dimensionality in high-dimensional conditioning sets $Z$, inherent in specific methods. Finally, the theoretical guarantees of E-CIT are conditional on the subtests being reasonably effective. As a divide-and-aggregate framework, E-CIT cannot substitute for the fundamental statistical validity of the underlying subtests.

*Practical Implementation.* Grounded in our theoretical analysis and empirical findings, we provide the following current guidelines for hyperparameter selection (see Appendix F for a detailed discussion). First, regarding the stability parameter $\alpha$, we recommend a general default setting of $\alpha \in \{1.75, 2\}$. While our framework offers the flexibility to tune $\alpha$ for optimality relative to specific base CITs, these values have demonstrated robustness across diverse settings in our experiments. Second, regarding the subset size $n_k$, it should be chosen such that the base CIT exhibits reasonable asymptotic behavior and power. A practical rule of thumb is to adopt the sample size recommended in the base CIT's original literature (e.g., $n_k = 400$ for KCIT (Zhang et al., 2011)).

*Limitations and Future Directions.* We further provide a detailed discussion on the limitations and future directions of our method in Appendix F. This includes an analysis of the potential impact on Type I error control, the handling of super-uniform p-values generated by permutation tests, and promising avenues for future research such as handling correlated p-values and developing method-specific enhancements. We believe E-CIT offers a powerful solution that balances computational efficiency with statistical power, thereby paving the way for causal discovery in large-scale and complex scientific problems.

### ACKNOWLEDGMENTS

This work was supported in part by the National Natural Science Foundation of China (62376243), the National Key Research and Development Program of China (2024YFE0203700), and "Pioneer" and "Leading Goose" R&D Program of Zhejiang (2025C02037).

A preliminary version of this work was presented as the bachelor's thesis of the first author, Zhengkang Guan. He would like to thank the faculty members of the Department of Statistics and Data Science at Xiamen University for their support and mentorship during his undergraduate studies. He is particularly grateful to Prof. Jingyuan Liu for her insightful discussions on general statistical methodologies, as well as her invaluable guidance regarding overall academic and career development.

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

## A   CLOSURE PROPERTY OF STABLE DISTRIBUTIONS

**Proposition A.1** (Nolan (2012)). *A stable distribution $\mathbf{S}(\alpha, \beta, \gamma, \delta)$ has the following properties:*

1. *If $X \sim \mathbf{S}(\alpha, \beta, \gamma, \delta)$, then for any $a \neq 0, b \in \mathbb{R}$,*

$$aX + b \sim \begin{cases} \mathbf{S}(\alpha, (\mathrm{sign}\, a)\beta, |a|\gamma, a\delta + b) & \alpha \neq 1 \\ \mathbf{S}\left(1, (\mathrm{sign}\, a)\beta, |a|\gamma, a\delta + b - \frac{2}{\pi}\beta\gamma a \log |a|\right) & \alpha = 1 \end{cases}$$

2. *If $X_1 \sim \mathbf{S}(\alpha, \beta_1, \gamma_1, \delta_1)$ and $X_2 \sim \mathbf{S}(\alpha, \beta_2, \gamma_2, \delta_2)$ are independent, then $X_1 + X_2 \sim \mathbf{S}(\alpha, \beta, \gamma, \delta)$, where*

$$\beta = \frac{\beta_1 \gamma_1^\alpha + \beta_2 \gamma_2^\alpha}{\gamma_1^\alpha + \gamma_2^\alpha}, \quad \gamma^\alpha = \gamma_1^\alpha + \gamma_2^\alpha, \quad \delta = \delta_1 + \delta_2$$

This is the general version of Proposition 1, which is one of the core properties of stable distributions. For detailed properties of stable distributions, refer to Nolan (2012; 2020).

## B   OMITTED PROOFS

### B.1   PROOF OF THEOREM 1

**Theorem 1.** *The ensemble test $\mathcal{H}_e$ (for exact subtest p-values) satisfies the following properties:*

1. *Validity: Under the null hypothesis, the ensemble p-value is uniformly distributed on $[0, 1]$, ensuring Type I error control.*

2. *Admissibility: The ensemble test is admissible, indicating that no other test uniformly outperforms it in terms of error rates and decision-making optimality.*

3. *Unbiasedness: The ensemble test is unbiased if its subtests are unbiased, meaning the ensemble does not compromise the unbiasedness of the individual subtests.*

*Proof.* We first establish *validity*, and then jointly prove *admissibility* and *unbiasedness*.

*Validity:*

According to the definition of the ensemble test and p-value (Definition 2), the validity property follows directly. First, by the definition of the p-value, under the null hypothesis, we have:

$$p_k \sim U(0, 1), \quad k = 1, \ldots, K.$$

Consider a stable distribution $S \sim \mathbf{S}(\alpha, \beta, \gamma, \delta)$ with cumulative distribution function (CDF) $F_S$, which is invertible, with its inverse denoted by $F_S^{-1}$. Therefore, we have:

$$P\left(F_S^{-1}(p_k) \leq s\right) = P\left(p_k \leq F_S(s)\right) = F_S(s).$$

Thus, we conclude

$$F_S^{-1}(p_k) \stackrel{d}{=} S \sim \mathbf{S}(\alpha, \beta, \gamma, \delta).$$

Furthermore, since $p_1, p_2, \ldots, p_K$ are derived from independent tests (and are thus i.i.d.), it follows that:

$$F_S^{-1}(p_1), F_S^{-1}(p_2), \ldots, F_S^{-1}(p_K) \stackrel{\text{i.i.d.}}{\sim} \mathbf{S}(\alpha, \beta, \gamma, \delta)$$

By Proposition 1, we obtain:

$$T_e = \frac{1}{K}\sum_{k=1}^{K} F_S^{-1}(p_k) \sim \mathbf{S}(\alpha, \beta, \gamma', \delta)$$

where $\gamma' = K^{\frac{1}{\alpha}-1}\gamma$.

Thus, by the Probability Integral Transform, it is evident that:

$$p_e = F_{S'}(T_e) \sim U(0, 1),$$

where $F_{S'}$ is the CDF of the stable distribution $\mathbf{S}(\alpha, \beta, \gamma', \delta)$.

***Admissibility and Unbiasedness:***

We build our proof on an earlier result presented by Lipták (1958), from which we formulate the following lemma.

**Lemma B.1** (Lipták (1958)). *Let $T_g$ be an aggregated statistic defined as*

$$T_g = x^{-1}\left(\sum_{i=1}^{K} w_i x(p_i)\right),$$

*where $x(\cdot)$ is any strictly increasing and continuous function, and $w_i$ represent weights satisfying $\sum_{i=1}^{K} w_i = 1$ and $w_i \in [0,1], i = 1, \ldots, K$. Then, the test based on $T_g$ is admissible. Furthermore, if the p-values $p_i, i = 1, \ldots, K$ are from unbiased tests, then the test based on $T_g$ is also unbiased.*

In Lemma B.1, the presence of the outer function $x^{-1}(\cdot)$ of $T_g$ yields an equivalent test, since $x(\cdot)$ is strictly increasing. By further setting equal weights and discarding constant terms, we derive the simplified statistic

$$T_g' = \sum_{i=1}^{K} x(p_i).$$

The test based on $T_g'$ also preserves both admissibility and unbiasedness.

Clearly, the ensemble test statistic

$$T_e = \frac{1}{K} \sum_{k=1}^{K} F_S^{-1}(p_k)$$

differs from this structure by only a constant factor. Therefore, the ensemble test also preserves admissibility and unbiasedness when $p_k, k = 1, \ldots, K$ are from unbiased subtests. $\qquad\square$

## B.2 Proof of Lemma 1

**Lemma 1.** *Assume that $F_S^{-1}(p_k^{H_1})$ is integrable. The power of the ensemble test $\mathcal{H}_e\,(\mathcal{H}, K; \alpha, \beta, \gamma, \delta)$ approaches 1 as $K \to \infty$, i.e., $\lim_{K \to \infty} \pi_e = 1$, if the following condition holds:*

$$\mathbb{E}\left[F_S^{-1}(p_k^{H_1})\right] < F_{S'}^{-1}(\alpha_e),$$

*where $p_k^{H_1}$ for $k = 1, 2, \ldots, K$ are i.i.d. p-values from the subtest $\mathcal{H}$ under the alternative hypothesis.*

*Proof.* By the definition of Type II error:

$$\begin{aligned}
\beta_e &= P\left(F_{S'}(T_e) > \alpha_e \mid H_1\right) \\
&= P\left(T_e > F_{S'}^{-1}(\alpha_e) \mid H_1\right) \\
&= P\left(\frac{1}{K} \sum_{k=1}^{K} F_S^{-1}(p_k^{H_1}) > F_{S'}^{-1}(\alpha_e)\right)
\end{aligned}$$

Since $F_S^{-1}(p_k^{H_1}), k = 1, 2, \ldots, K$ are i.i.d. and integrable, by the Strong Law of Large Numbers (SLLN), we have:

$$T_e = \frac{1}{K} \sum_{k=1}^{K} F_S^{-1}(p_k^{H_1}) \xrightarrow{a.s.} \mathbb{E}\left[F_S^{-1}(p_k^{H_1})\right] \quad \text{as } K \to \infty.$$

Then for any $\epsilon > 0$, there exists (almost surely) $K_0$ such that for all $K \geq K_0$:

$$T_e < \mathbb{E}\left[F_S^{-1}(p_k^{H_1})\right] + \epsilon.$$

Take $\epsilon = F_{S'}^{-1}(\alpha_e) - \mathbb{E}\left[F_S^{-1}(p_k^{H_1})\right] > 0$ since we have $\mathbb{E}\left[F_S^{-1}(p_k^{H_1})\right] < F_{S'}^{-1}(\alpha_e)$, then almost surely for large enough $K$:

$$T_e < F_{S'}^{-1}(\alpha_e).$$

Therefore:

$$P\left(T_e > F_{S'}^{-1}(\alpha_e)\right) \to 0 \quad \text{as } K \to \infty,$$

which implies:

$$\lim_{K \to \infty} \beta_e = 0.$$

This is equivalent to:

$$\lim_{K \to \infty} \pi_e = 1.$$

$\square$

### B.3 PROOF OF THEOREM 2

**Theorem 2.** *Consider the ensemble test $\mathcal{H}_e\left(\mathcal{H}, K; \alpha, \beta, \gamma, \delta\right)$, and assume that $F_S^{-1}(p_k^{H_1})$ is integrable. If the following conditions hold:*

1. $\mathbb{E}[p_k^{H_1}] \leq \alpha_e$,

2. $f_1(p) \geq f_1(1-p)$ for $p \in \left[0, \frac{1}{2}\right]$, where $f_1$ is the probability density function of $p_k^{H_1}$,

3. $\alpha \geq 1, \beta = \delta = 0$.

*Then, we have $\lim_{K \to \infty} \pi_e = 1$.*

*Proof.* It is sufficient to show that under these conditions, we can derive

$$\mathbb{E}\left[F_S^{-1}(p_k^{H_1})\right] \leq F_{S'}^{-1}(\alpha_e)$$

which directly yields the conclusion via Lemma 1.

The first step is to show that $\mathbb{E}\left[F_S^{-1}(p_k^{H_1})\right] < F_S^{-1}\left(\mathbb{E}\left[p_k^{H_1}\right]\right)$:

We begin by reformulating $\mathbb{E}\left[F_S^{-1}(p_k^{H_1})\right]$ to facilitate bounding:

$$\begin{aligned}
\mathbb{E}\left[F_S^{-1}(p_k^{H_1})\right] &= \int_0^1 F_S^{-1}(p) \cdot f_1(p) \, dp \\
&= \int_0^{\frac{1}{2}} F_S^{-1}(p) \cdot f_1(p) \, dp + \int_{\frac{1}{2}}^1 F_S^{-1}(p) \cdot f_1(p) \, dp \\
&= \int_0^{\frac{1}{2}} F_S^{-1}(p) \cdot f_1(p) \, dp - \int_0^{\frac{1}{2}} F_S^{-1}(p) \cdot f_1(1-p) \, dp \\
&= \int_0^{\frac{1}{2}} F_S^{-1}(p) \cdot [f_1(p) - f_1(1-p)] \, dp
\end{aligned}$$

Consider the tangent line $l(\cdot)$ of $F_S^{-1}$ at $\mathbb{E}\left[p_k^{H_1}\right]$. Since $\beta = \delta = 0$, $F_S$ is monotonically increasing and convex on $\left[0, \frac{1}{2}\right]$, and therefore $F_S^{-1}$ is monotonically increasing and concave on $\left[0, \frac{1}{2}\right]$. Additionally, since $\mathbb{E}\left[p_k^{H_1}\right] \leq \alpha_e \leq \frac{1}{2}$, it follows that $l(p) \geq F_S^{-1}(p)$ for $p \in \left[0, \frac{1}{2}\right]$, and $l(\cdot)$ is also monotonically increasing.

Meanwhile, since $f_1(p) \geq f_1(1-p)$ for $p \in \left[0, \frac{1}{2}\right]$, we have $f_1(p) - f_1(1-p) \geq 0$ for $p \in \left[0, \frac{1}{2}\right]$.

Therefore, we obtain:

$$\int_0^{\frac{1}{2}} F_S^{-1}(p) \cdot [f_1(p) - f_1(1-p)] \, dp \le \int_0^{\frac{1}{2}} l(p) \cdot [f_1(p) - f_1(1-p)] \, dp$$

$$= \int_0^{\frac{1}{2}} l(p) \cdot f_1(p) \, dp - \int_0^{\frac{1}{2}} l(p) \cdot f_1(1-p) \, dp$$

$$= \int_0^{\frac{1}{2}} l(p) \cdot f_1(p) \, dp + \int_{\frac{1}{2}}^1 -l(1-p) \cdot f_1(p) \, dp$$

Since $\beta = \delta = 0$, it follows from the definition of the stable distribution that $F_S^{-1}\left(\frac{1}{2}\right) = 0$ holds. Furthermore, as $l(\cdot)$ is the tangent line of $F_S^{-1}$ at $\mathbb{E}\left[p_k^{H_1}\right]$ and $F_S^{-1}$ is concave on $\left[0, \frac{1}{2}\right]$, we have

$$l\left(\frac{1}{2}\right) > F_S^{-1}\left(\frac{1}{2}\right) = 0 > -l\left(\frac{1}{2}\right).$$

Moreover, it follows that $l(p)$ and $-l(1-p)$ are parallel, which implies that $l(p) > -l(1-p)$ for any $p$. Consequently, we obtain

$$\int_0^{\frac{1}{2}} l(p) \cdot f_1(p) \, dp + \int_{\frac{1}{2}}^1 -l(1-p) \cdot f_1(p) \, dp < \int_0^{\frac{1}{2}} l(p) \cdot f_1(p) \, dp + \int_{\frac{1}{2}}^1 l(p) \cdot f_1(p) \, dp$$

$$= \int_0^1 l(p) \cdot f_1(p) \, dp$$

$$= \mathbb{E}\left[l(p_k^{H_1})\right].$$

Because $l(\cdot)$ is linear and tangent to $F_S^{-1}$ at $\mathbb{E}[p_k^{H_1}]$,

$$\mathbb{E}\left[l(p_k^{H_1})\right] = l\left(\mathbb{E}\left[p_k^{H_1}\right]\right)$$

$$= F_S^{-1}\left(\mathbb{E}\left[p_k^{H_1}\right]\right).$$

Thus, we obtain

$$\mathbb{E}\left[F_S^{-1}(p_k^{H_1})\right] < F_S^{-1}\left(\mathbb{E}\left[p_k^{H_1}\right]\right). \tag{1}$$

Since $F_S^{-1}$ is monotonically increasing and $\mathbb{E}\left[p_k^{H_1}\right] \le \alpha_e$, we further deduce that

$$F_S^{-1}\left(\mathbb{E}\left[p_k^{H_1}\right]\right) \le F_S^{-1}\left(\alpha_e\right). \tag{2}$$

Moreover, given that $\alpha \ge 1$, we have $\gamma' = K^{\frac{1}{\alpha}-1}\gamma \le \gamma$, indicating that $\mathbf{S}(\alpha, \beta, \gamma', \delta)$ has a smaller scale parameter compared to $\mathbf{S}(\alpha, \beta, \gamma, \delta)$. Furthermore, when $\beta = \delta = 0$, both distributions are symmetric, and it is evident that:

$$F_S^{-1}\left(\alpha_e\right) \le F_{S'}^{-1}\left(\alpha_e\right). \tag{3}$$

Combining inequalities (1), (2), and (3), we conclude that

$$\mathbb{E}\left[F_S^{-1}(p_k^{H_1})\right] < F_{S'}^{-1}\left(\alpha_e\right).$$

By applying Lemma 1, we obtain $\lim_{K \to \infty} \pi_e = 1$. $\qquad\square$

## C    PRACTICAL CONSIDERATIONS OF CERTAIN CONSISTENCY GUARANTEES

While some CIT methods offer theoretical consistency guarantees, it is important to note that their practical performance can be compromised by the challenging nature of real-world data. We acknowledge the asymptotic guarantees, but highlight how complex data environments can limit their practical applicability. It has been shown that no single CIT can be effective in all scenarios (Shah & Peters, 2018; Kim et al., 2021), which further implies that universal consistency across all scenarios is unattainable for any single CIT.

To illustrate these practical limitations, we consider LPCIT (Scetbon et al., 2022) as an example, focusing on how its assumptions and estimation procedures can affect its convergence speed in practice:

- Assumption Violations: LPCIT's consistency derivation relies on Assumption 3.5 (Scetbon et al., 2022), which requires the variables under test, after kernel mapping, to possess higher-order moments with controlled growth rates. However, in many practical situations, the variables might be heavy-tailed, leading to heavy-tailed properties even after kernel mapping. This can violate the assumption, potentially causing a failure of consistency in practice. Experiments in Section 4.2 using $t$-distributions with varying tail thicknesses also show that LPCIT performs better in relatively thin-tailed scenarios.

- Estimation Difficulties: LPCIT's estimation of conditional means relies on Regularized Least Squares (RLS), which minimizes squared error and is highly sensitive to extreme values. This implies that as the sample size increases, extreme values can disproportionately affect the squared error term, making it difficult for the estimator's variance to effectively decrease, thus limiting the improvement in test performance.

- Hyperparameter Optimization Challenges: LPCIT employs Gaussian process regression for selecting kernel bandwidth and RLS regularization parameters. In our experiments, we found that this optimization process in LPCIT is highly non-convex. The complexity of CIT scenarios makes it challenging to perfectly solve for the aforementioned hyperparameters, which may further limit the power improvement as the sample size increases.

## D    CERTAIN ILLUSTRATIONS OF THE CONDITIONS IN THEOREM 2

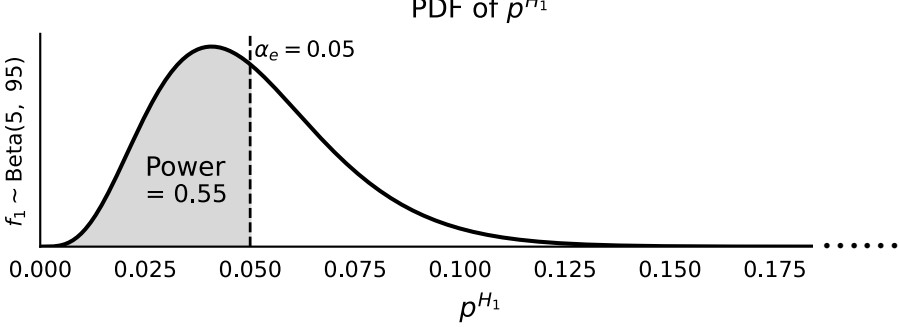

Figure 4: An Example Satisfying the First Two Conditions of Theorem 2: $p_k^{H_1} \sim \text{Beta}(5, 95)$

Consider the first two conditions of Theorem 2 for a subtest $\mathcal{H}$:

1. $\mathbb{E}\left[p_k^{H_1}\right] \leq \alpha_e$,

2. $f_1(p) \geq f_1(1 - p)$ for $p \in \left[0, \frac{1}{2}\right]$, where $f_1$ is the probability density function of $p_k^{H_1}$.

Here we consider modeling the distribution of $p_k^{H_1}$ using the Beta distribution (Heard & Rubin-Delanchy, 2018). Figure 4 illustrates an idealized case where the significance level is set to 0.05 and $p_k^{H_1} \sim \text{Beta}(5, 95)$, so that its expectation equals 0.05, which satisfies the first condition of

Theorem 2. In this setting, the power of the test corresponds to the probability that $p_k^{H_1} < 0.05$, meaning the probability of correctly rejecting the null hypothesis under the alternative. As shown by the shaded area in Figure 4, this probability is approximately 0.55. In more extreme cases, the area can approach 0.5. This indicates that the first condition of Theorem 2 can be seen as a requirement on the power of the subtest under $f_1$, and this requirement is relatively mild.

Next, we demonstrate that when $p_k^{H_1}$ follows a Beta distribution, the second condition naturally follows from the first.

Assuming $p_k^{H_1}$ follows a Beta distribution, its probability density function is:

$$f_1(p; \alpha_B, \beta_B) = \frac{1}{B(\alpha_B, \beta_B)} p^{\alpha_B - 1} (1 - p)^{\beta_B - 1}, \quad 0 < p < 1$$

From the first condition of Theorem 2, we have:

$$\mathbb{E}[p] = \frac{\alpha_B}{\alpha_B + \beta_B} \leq \alpha_e.$$

Thus, we have

$$\frac{\beta_B}{\alpha_B} \geq \frac{1}{\alpha_e} - 1 \geq 1.$$

For $p \in \left[0, \frac{1}{2}\right]$, taking the ratio gives:

$$\frac{f_1(p; \alpha_B, \beta_B)}{f_1(1 - p; \alpha_B, \beta_B)} = \left(\frac{p}{1 - p}\right)^{\alpha_B - \beta_B}$$

Since $\alpha_B - \beta_B < 0$ and $\frac{p}{1-p} \leq 1$ for $p \in \left[0, \frac{1}{2}\right]$, we have $\left(\frac{p}{1-p}\right)^{\alpha_B - \beta_B} \geq 1$. Therefore:

$$f_1(p) \geq f_1(1 - p), \quad \forall p \in \left[0, \frac{1}{2}\right]$$

which is the second condition of Theorem 2.

## E    ADDITIONAL EXPERIMENTS RESULTS

### E.1    EMPIRICAL STUDY ON THE SELECTION OF $\alpha$

We investigate how the parameter $\alpha$ affects the performance of E-CIT, under a post-nonlinear model similar to Section 4.1. We use standard Laplace-distributed noise and normally distributed $Z$, with $n = 1200$. Ensemble KCIT (E-KCIT) with $n_k = 400$ is used as a representative, with $\alpha \in \{0.25, 0.5, \ldots, 2\}$. As a baseline, we also include a mean-p method that directly averages p-values.

Figure 5 shows that the power of E-KCIT increases with larger $\alpha$, consistent with Theorem 2, while Type I error follows a non-monotonic trend. Overall, $\alpha = 1.75$ and 2 yield the best performance under this setting and are used in subsequent experiments under different data scenarios.

### E.2    ADDITIONAL RESULTS FOR SECTION 4.2 EXPERIMENTS

In the main text, we present the representative Table 2, which reports results for the setting with $n = 1200$ and $Z$ following the standard normal distribution. The results under other data settings are shown in the following tables.

In addition, we note that the experiments in this section are conducted using $t$-distributions with different degrees of freedom. The rationale is that some methods behave uncontrollably under extreme distributions such as the Cauchy, while the Gaussian distribution and the Laplace distribution pose little challenge for many methods. Therefore, we choose $t$-distributions with varying degrees of freedom to ensure both reasonable and diverse comparisons.

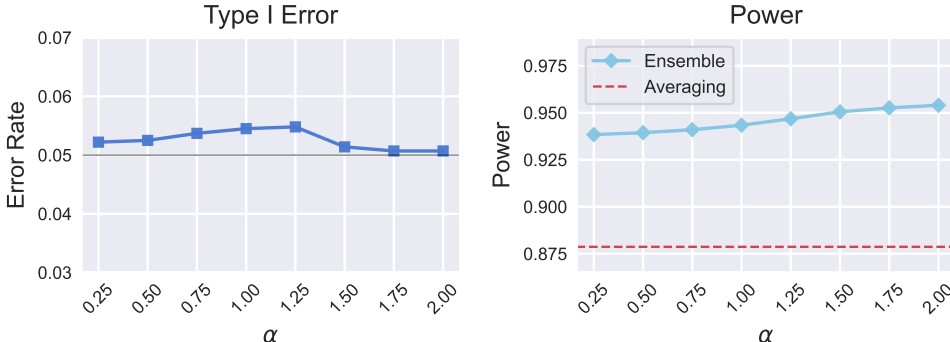

Figure 5: Empirical evaluation of $\alpha$ using E-KCIT. Type I error (left) and power (right). The red line indicates the power of mean-p.

Table 4: Results for $n = 800$, Standard Normal $Z$. Bold (*Bold italics*) indicates the ensemble (original) version is statistically significantly better.

| df of $t$-distributed Noise | | df = 4 | | df = 3 | | df = 2 | |
|---|---|---|---|---|---|---|---|
| | CIT Methods | Type I | Power | Type I | Power | Type I | Power |
| RCIT | Orig. | 0.037 | 0.801 | 0.041 | 0.724 | 0.042 | 0.512 |
| | Ensemble $_{(\alpha=1.75)}$ | 0.041 | **0.833** | 0.039 | **0.768** | 0.033 | **0.545** |
| | Ensemble $_{(\alpha=2.00)}$ | 0.060 | **0.852** | 0.048 | **0.786** | 0.045 | **0.562** |
| LPCIT | Orig. | 0.065 | 0.740 | 0.052 | 0.627 | 0.016 | 0.423 |
| | Ensemble $_{(\alpha=1.75)}$ | 0.051 | 0.745 | 0.030 | **0.681** | 0.021 | 0.436 |
| | Ensemble $_{(\alpha=2.00)}$ | 0.045 | 0.720 | 0.033 | **0.662** | 0.015 | 0.423 |
| CMIknn | Orig. | 0.086 | 0.978 | 0.124 | 0.984 | 0.106 | 0.954 |
| | Ensemble $_{(\alpha=1.75)}$ | 0.110 | 0.974 | 0.104 | 0.978 | 0.128 | 0.958 |
| | Ensemble $_{(\alpha=2.00)}$ | 0.088 | 0.976 | 0.116 | 0.972 | 0.108 | 0.964 |
| CCIT | Orig. | 0.414 | *0.904* | 0.426 | *0.886* | 0.414 | *0.882* |
| | Ensemble $_{(\alpha=1.75)}$ | **0.372** | 0.838 | **0.358** | 0.844 | **0.352** | 0.816 |
| | Ensemble $_{(\alpha=2.00)}$ | 0.400 | 0.844 | 0.424 | 0.848 | 0.392 | 0.814 |
| FisherZ | Orig. | 0.170 | 0.702 | 0.129 | 0.585 | 0.090 | 0.502 |
| | Ensemble $_{(\alpha=1.75)}$ | 0.172 | 0.702 | 0.125 | **0.625** | 0.081 | 0.520 |
| | Ensemble $_{(\alpha=2.00)}$ | 0.185 | 0.683 | 0.109 | **0.636** | 0.066 | **0.532** |

Table 5: Results for $n = 800$, Standard Laplace $Z$. Bold (*Bold italics*) indicates the ensemble (original) version is statistically significantly better.

| df of $t$-distributed Noise | | df = 4 | | df = 3 | | df = 2 | |
|---|---|---|---|---|---|---|---|
| CIT Methods | | Type I | Power | Type I | Power | Type I | Power |
| RCIT | Orig. | 0.043 | 0.801 | 0.056 | 0.688 | *0.019* | 0.521 |
| | Ensemble $(\alpha=1.75)$ | 0.046 | 0.803 | 0.048 | **0.720** | 0.045 | 0.536 |
| | Ensemble $(\alpha=2.00)$ | 0.058 | 0.796 | 0.051 | **0.745** | 0.047 | **0.552** |
| LPCIT | Orig. | 0.055 | 0.714 | 0.042 | 0.636 | 0.017 | 0.422 |
| | Ensemble $(\alpha=1.75)$ | 0.050 | 0.719 | 0.037 | 0.637 | 0.019 | 0.434 |
| | Ensemble $(\alpha=2.00)$ | 0.046 | 0.730 | 0.039 | 0.647 | 0.021 | **0.449** |
| CMIknn | Orig. | 0.122 | 0.960 | 0.114 | 0.956 | *0.090* | 0.938 |
| | Ensemble $(\alpha=1.75)$ | 0.116 | 0.962 | 0.130 | 0.962 | 0.118 | 0.956 |
| | Ensemble $(\alpha=2.00)$ | 0.120 | 0.970 | 0.126 | 0.952 | 0.122 | 0.954 |
| CCIT | Orig. | 0.424 | 0.874 | 0.434 | *0.882* | 0.430 | *0.850* |
| | Ensemble $(\alpha=1.75)$ | **0.332** | 0.862 | **0.356** | 0.806 | **0.378** | 0.818 |
| | Ensemble $(\alpha=2.00)$ | **0.386** | 0.828 | **0.386** | 0.832 | 0.418 | 0.824 |
| FisherZ | Orig. | 0.476 | 0.688 | 0.274 | 0.645 | 0.118 | 0.525 |
| | Ensemble $(\alpha=1.75)$ | 0.484 | 0.682 | 0.320 | 0.648 | 0.127 | **0.550** |
| | Ensemble $(\alpha=2.00)$ | 0.459 | 0.667 | 0.299 | 0.630 | 0.119 | **0.594** |

Table 6: Results for $n = 1200$, Standard Laplace $Z$. Bold (*Bold italics*) indicates the ensemble (original) version is statistically significantly better.

| df of $t$-distributed Noise | | df = 4 | | df = 3 | | df = 2 | |
|---|---|---|---|---|---|---|---|
| CIT Methods | | Type I | Power | Type I | Power | Type I | Power |
| RCIT | Orig. | 0.049 | 0.824 | 0.046 | 0.730 | 0.041 | 0.532 |
| | Ensemble $(\alpha=1.75)$ | 0.067 | **0.860** | 0.053 | **0.792** | 0.040 | **0.578** |
| | Ensemble $(\alpha=2.00)$ | 0.060 | **0.869** | 0.068 | **0.822** | 0.040 | **0.622** |
| LPCIT | Orig. | 0.065 | 0.762 | 0.034 | 0.631 | 0.028 | 0.430 |
| | Ensemble $(\alpha=1.75)$ | 0.039 | 0.755 | 0.034 | **0.690** | 0.013 | **0.464** |
| | Ensemble $(\alpha=2.00)$ | 0.050 | 0.747 | 0.041 | **0.678** | 0.015 | 0.436 |
| CMIknn | Orig. | 0.100 | 0.992 | 0.108 | 0.968 | 0.084 | 0.960 |
| | Ensemble $(\alpha=1.75)$ | 0.150 | 0.980 | 0.136 | 0.982 | 0.130 | 0.982 |
| | Ensemble $(\alpha=2.00)$ | 0.082 | 0.982 | 0.128 | 0.972 | 0.098 | 0.984 |
| CCIT | Orig. | 0.442 | *0.898* | 0.428 | *0.900* | 0.458 | *0.892* |
| | Ensemble $(\alpha=1.75)$ | **0.318** | 0.806 | **0.296** | 0.816 | **0.326** | 0.788 |
| | Ensemble $(\alpha=2.00)$ | **0.348** | 0.862 | **0.318** | 0.810 | **0.312** | 0.812 |
| FisherZ | Orig. | 0.484 | 0.691 | *0.287* | 0.682 | 0.138 | 0.520 |
| | Ensemble $(\alpha=1.75)$ | 0.545 | **0.726** | 0.371 | 0.685 | 0.134 | **0.591** |
| | Ensemble $(\alpha=2.00)$ | 0.489 | **0.731** | 0.332 | 0.692 | 0.136 | **0.577** |

Table 7: Results for $n = 1600$, Standard Normal $Z$. Bold (*Bold italics*) indicates the ensemble (original) version is statistically significantly better.

| df of $t$-distributed Noise | | df = 4 | | df = 3 | | df = 2 | |
|---|---|---|---|---|---|---|---|
| CIT Methods | | Type I | Power | Type I | Power | Type I | Power |
| RCIT | Orig. | 0.039 | 0.866 | 0.042 | 0.799 | 0.037 | 0.551 |
| | Ensemble $(\alpha=1.75)$ | 0.048 | **0.932** | 0.047 | **0.870** | 0.043 | **0.642** |
| | Ensemble $(\alpha=2.00)$ | 0.069 | **0.931** | 0.069 | **0.883** | 0.065 | **0.681** |
| LPCIT | Orig. | 0.072 | 0.744 | 0.055 | 0.674 | 0.024 | 0.415 |
| | Ensemble $(\alpha=1.75)$ | **0.035** | 0.755 | **0.023** | **0.707** | 0.012 | **0.465** |
| | Ensemble $(\alpha=2.00)$ | 0.053 | 0.754 | **0.023** | 0.696 | 0.015 | 0.423 |
| CMIknn | Orig. | *0.102* | 0.994 | 0.104 | 0.996 | 0.114 | 0.994 |
| | Ensemble $(\alpha=1.75)$ | 0.146 | 0.998 | 0.128 | 0.998 | 0.164 | 0.986 |
| | Ensemble $(\alpha=2.00)$ | 0.142 | 1.000 | 0.104 | 0.996 | 0.130 | 1.000 |
| CCIT | Orig. | 0.428 | *0.932* | 0.434 | *0.928* | 0.428 | *0.926* |
| | Ensemble $(\alpha=1.75)$ | **0.232** | 0.834 | **0.256** | 0.806 | **0.230** | 0.752 |
| | Ensemble $(\alpha=2.00)$ | **0.270** | 0.828 | **0.260** | 0.826 | **0.260** | 0.802 |
| FisherZ | Orig. | 0.232 | 0.756 | 0.130 | 0.634 | 0.105 | 0.526 |
| | Ensemble $(\alpha=1.75)$ | 0.248 | 0.759 | 0.142 | **0.672** | 0.112 | **0.576** |
| | Ensemble $(\alpha=2.00)$ | 0.216 | 0.744 | 0.119 | **0.665** | 0.101 | 0.535 |

Table 8: Results for $n = 1600$, Standard Laplace $Z$. Bold (*Bold italics*) indicates the ensemble (original) version is statistically significantly better.

| df of $t$-distributed Noise | | df = 4 | | df = 3 | | df = 2 | |
|---|---|---|---|---|---|---|---|
| CIT Methods | | Type I | Power | Type I | Power | Type I | Power |
| RCIT | Orig. | 0.046 | 0.862 | 0.043 | 0.757 | 0.038 | 0.553 |
| | Ensemble $(\alpha=1.75)$ | 0.062 | **0.913** | 0.053 | **0.849** | 0.039 | **0.639** |
| | Ensemble $(\alpha=2.00)$ | 0.071 | **0.916** | 0.060 | **0.882** | 0.060 | **0.673** |
| LPCIT | Orig. | 0.075 | 0.755 | 0.062 | 0.663 | 0.018 | 0.421 |
| | Ensemble $(\alpha=1.75)$ | **0.043** | 0.750 | **0.037** | **0.716** | 0.009 | **0.482** |
| | Ensemble $(\alpha=2.00)$ | **0.045** | 0.772 | 0.040 | **0.714** | 0.014 | **0.453** |
| CMIknn | Orig. | *0.098* | 0.990 | 0.112 | 0.980 | 0.114 | 0.982 |
| | Ensemble $(\alpha=1.75)$ | 0.146 | 0.992 | 0.160 | 0.994 | 0.144 | 0.994 |
| | Ensemble $(\alpha=2.00)$ | 0.116 | 0.994 | 0.134 | 0.990 | 0.106 | 0.994 |
| CCIT | Orig. | 0.452 | *0.936* | 0.462 | *0.910* | 0.436 | *0.908* |
| | Ensemble $(\alpha=1.75)$ | **0.266** | 0.810 | **0.230** | 0.788 | **0.254** | 0.768 |
| | Ensemble $(\alpha=2.00)$ | **0.282** | 0.788 | **0.248** | 0.818 | **0.280** | 0.758 |
| FisherZ | Orig. | *0.509* | 0.707 | 0.337 | 0.682 | *0.121* | 0.525 |
| | Ensemble $(\alpha=1.75)$ | 0.554 | **0.747** | 0.375 | 0.703 | 0.164 | **0.558** |
| | Ensemble $(\alpha=2.00)$ | 0.556 | 0.720 | 0.361 | 0.704 | 0.157 | **0.612** |

### E.3 Impact of Dimensionality of Conditioning Set $Z$

Figure 6 illustrates the effect of conditioning set $Z$ dimensionality on the performance of KCIT, RCIT, LPCIT, and Fisher Z-test, using a fixed sample size of 1200, with $Z$ sampled from a standard Gaussian and noise from a $t$-distribution with df=3. For each method, we plot the Type I error rate (left subfigure) and the test power (right subfigure) as a function of the dimensionality of $Z$.

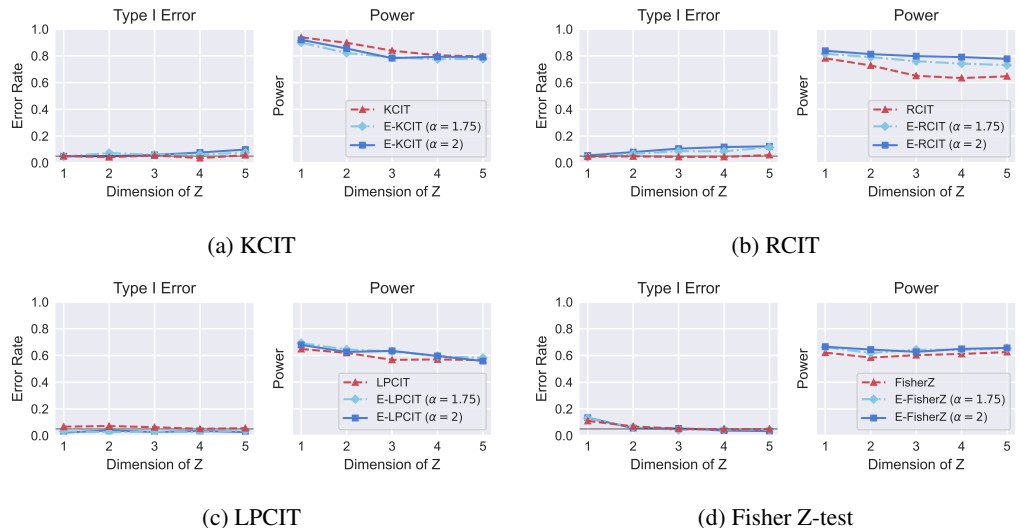

Figure 6: Effect of the conditioning set dimensionality on the ensemble performance of KCIT, RCIT, LPCIT, and Fisher Z-test.

Although E-KCIT does not perform well at this relatively small sample size, the ensemble framework generally provides performance improvements across methods and conditioning set dimensionalities, further demonstrating its effectiveness for CIT.

### E.4 DETAILS OF EXPERIMENT ON REAL-WORLD FLOW-CYTOMETRY DATA

We conduct experiments on the Flow-Cytometry dataset, a widely used benchmark for evaluating CIT methods and causal discovery algorithms (Li et al., 2023b; Sen et al., 2017; Mukherjee et al., 2020; Ng et al., 2020; Zhu et al., 2019; Mooij & Heskes, 2013). This dataset originates from the seminal study by Sachs et al. (2005), which employed multiparameter flow cytometry to measure single-cell signaling in primary human CD4+ T cells. The dataset is available in the supplementary material of Sachs et al. (2005). (It is important to clarify that this dataset consists of real-world biological data, not computer simulations. The term "stimulations" in the original study refers to biological reagents used to perturb the cells, not synthetic data generation.)

The dataset comprises measurements of 11 phosphorylated proteins and phospholipids. The consensus causal graph, derived by domain experts and validated in the original study, serves as the ground truth. Although the data were collected under biological stimulations, we follow the standard protocol in CIT literature (Li et al., 2023b; Sen et al., 2017; Mukherjee et al., 2020; Ng et al., 2020; Zhu et al., 2019; Mooij & Heskes, 2013) by treating specific subsets as observational data. This is justified because the stimulations act on unobserved exogenous root nodes, preserving the downstream causal structure among the observed proteins.

To provide a comprehensive evaluation and address different sample size regimes, we utilize two configurations of this dataset:

- **Dataset A ($n = 1755$):** Following the setup of Li et al. (2023b), we combine the samples from the `cd3cd28` and `cd3cd28icam2` experimental conditions. This results in a total of 1755 samples. This setting tests the methods' performance on a moderate sample size.

- **Dataset B ($n = 853$):** To align with benchmarks such as CCMI (Mukherjee et al., 2020) and evaluate performance on smaller sample sizes, we explicitly evaluate on the `cd3cd28` alone, which contains 853 samples.

From the consensus graph, we extract a comprehensive set of 50 conditionally independent and 50 conditionally dependent pairs for evaluation. Similar to Section 4.2, we compare six CIT methods and their two ensemble variants. We evaluate performance using precision, recall, and F1-score.

Table 9: Performance comparison on the Flow-Cytometry dataset B (results for both $\alpha$ merged). Bold (*Bold italics*) indicates the ensemble (original) version is statistically significantly better.

| Method | | Precision | Recall | F1-score |
|---|---|---|---|---|
| KCIT | Orig. | ***0.880*** | 0.667 | ***0.759*** |
| | Ensemble | 0.862 | 0.660 | 0.747 |
| RCIT | Orig. | ***0.888*** | 0.654 | 0.753 |
| | Ensemble | 0.872 | **0.665** | 0.754 |
| LPCIT | Orig. | 0.912 | 0.660 | 0.766 |
| | Ensemble | **0.924** | **0.673** | **0.778** |
| CMIknn | Orig. | ***0.920*** | 0.667 | 0.773 |
| | Ensemble | 0.904 | 0.667 | 0.767 |
| CCIT | Orig. | 0.560 | 0.636 | 0.596 |
| | Ensemble | **0.594** | **0.658** | **0.623** |
| FisherZ | Orig. | 0.940 | 0.671 | 0.783 |
| | Ensemble | 0.942 | 0.672 | 0.784 |

Precision and recall are defined as $TP/(TP + FP)$ and $TP/(TP + FN)$, respectively. The F1-score is the harmonic mean of the two. Here, TP and TN denote correctly identified conditionally dependent and independent instances, respectively, while FP and FN denote incorrect predictions.

Consistent with our hyperparameter guideline to maintain a subset size $n_k \approx 400$, we set $K = 5$ for Dataset A ($n_k \approx 351$) and $K = 2$ for Dataset B ($n_k \approx 426$). We evaluate E-CIT with both $\alpha = 1.75$ and $\alpha = 2$. As the results were stable across these values, we report them jointly. Due to the randomness in data partitioning, E-CIT results are averaged over 10 runs. RCIT (Strobl et al., 2019) and its ensemble version are averaged over 100 runs to account for their inherent randomness. The results for the two datasets are presented in Tables 3 and 9.

### E.5  DETAILS OF EXPERIMENT ON THE APPLICATION IN CAUSAL DISCOVERY

In the causal discovery experiments, we generate synthetic causal graphs as follows. Each graph contains a backbone path generated according to a fixed topological order, while all other possible edges are added independently with probability 0.3. Data are then simulated according to the graph structure: each variable is computed as the sum of its parent variables after transformation by a nonlinear function (randomly chosen from $\{x, x^2, x^3, \tanh(x), \cos(x)\}$), with additive noise drawn from a standard Student's $t$ (df = 2), Laplace, or Cauchy distribution. We apply the PC algorithm (Spirtes et al., 2000) 50 times using each CIT method.

We exclude FastKCIT (Schacht & Huang, 2025) from this comparison because its assumption that the conditioning set can be well approximated by a Gaussian mixture with $V$ components results in highly unstable performance in strongly non-Gaussian scenarios. Figure 3 reports results in terms of F1-score (left), SHD (middle), and runtime (right). SHD measures the number of edge operations required to transform the estimated graph into the ground-truth graph. To isolate the effect of conditional independence testing from that of edge orientation rules in the PC algorithm, both F1-score and SHD are computed on skeleton graphs only.

### E.6  ABLATION STUDY OF SUBSET SIZE

We investigate how the choice of the subset size $n_k$ affects the performance of E-CIT under the same post-nonlinear model setup as in Section 4.1. Specifically, we fix the total sample size to $n = 2000$ and consider $n_k \in \{200, 285, 333, 400, 500, 666, 1000\}$. For each configuration, we conduct 1000 independent experiments and compare the performance of E-KCIT (with $\alpha = 1.75$ and $\alpha = 2$) against the original KCIT. Results are summarized in Figure 7.

Overall, although the subset size $n_k$ has some impact on E-KCIT's performance, the ensemble method shows competitive performance relative to the original KCIT across most values, demonstrating the

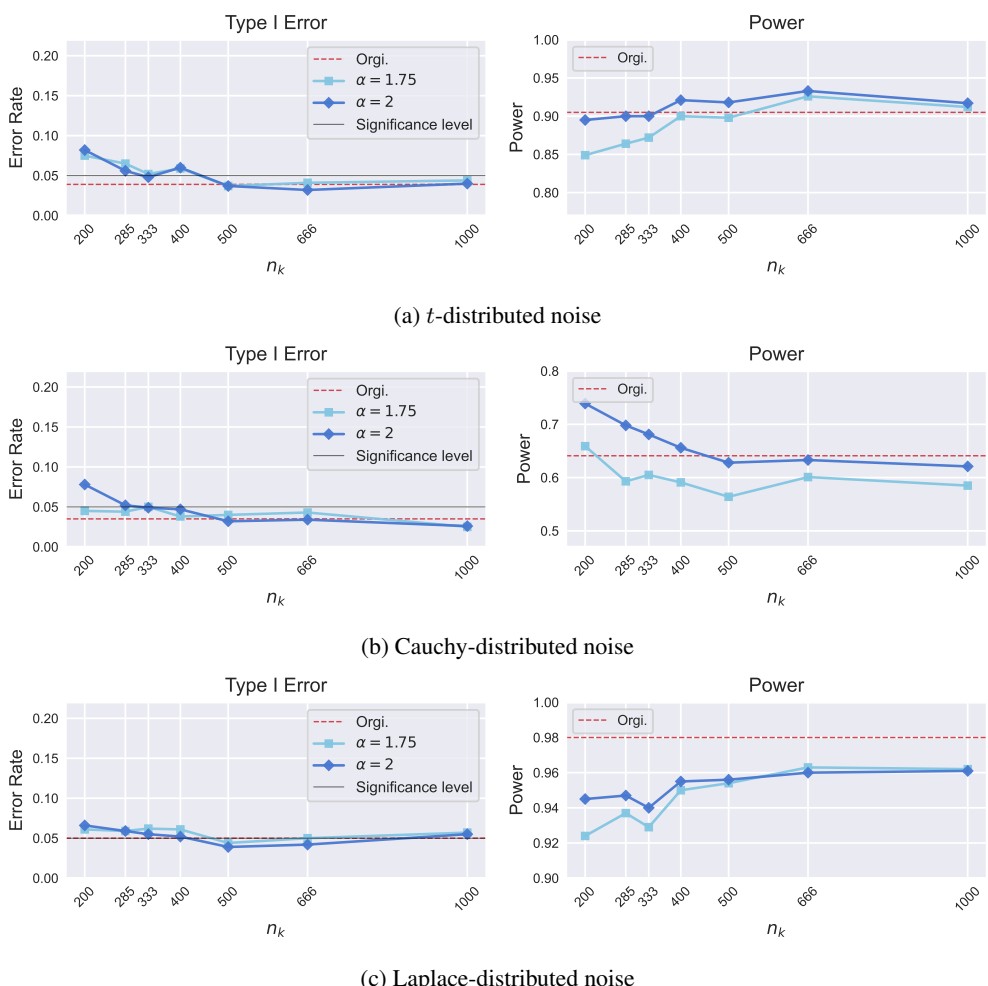

(a) $t$-distributed noise

(b) Cauchy-distributed noise

(c) Laplace-distributed noise

Figure 7: Comparison of Type I error (left), test power (right) for E-KCIT (with different $n_k$) and KCIT under different noise distributions.

robustness of E-CIT with respect to this parameter. Importantly, the choice of $n_k = 400$ adopted in our other experiments (an empirical choice following previous CIT studies) is not necessarily optimal. This observation highlights the potential for further gains within our framework. Nevertheless, we note that $n_k$ can influence the control of Type I error. This arises because the asymptotic distribution of subtest p-values may deviate from Uniform$(0, 1)$ depending on $n_k$. As shown in Figure 7, this effect is mild and can be mitigated by simply avoiding excessively small values of $n_k$ to ensure the asymptotic distribution remains close to Uniform$(0, 1)$. For a more detailed discussion, please refer to Appendix F.

### E.7 P-VALUE COMBINATION METHODS FOR E-CIT

We further compare our proposed p-value combination method with several classical alternatives. We fix the total sample size at $n = 2000$ and subset size at $n_k = 400$, while keeping all other settings identical to those in Section 4.1. We perform 1000 repetitions of E-KCIT under three different noise distributions, and report results in Table 10.

Across all scenarios, our method achieves the highest power while maintaining valid Type I error control, consistently outperforming the classical alternatives. It is worth noting that when $\alpha = 2$, our method reduces to Stouffer et al. (1949). In this case, we omit it from the table.

Table 10: Comparison of different combination methods

| Noise | Metric | Combination Methods | | | | | |
|---|---|---|---|---|---|---|---|
| | | Ours | Tippett | Edgington | Fisher | Pearson | Mudholkar |
| $t$ | Type I | 0.033 | 0.035 | *0.289* | 0.035 | *0.096* | 0.018 |
| | Power | **0.919** | 0.735 | *0.956* | 0.876 | *0.933* | 0.876 |
| Cauchy | Type I | 0.032 | 0.031 | *0.463* | 0.028 | *0.196* | 0.015 |
| | Power | **0.649** | 0.467 | *0.925* | 0.546 | *0.821* | 0.547 |
| Laplace | Type I | 0.065 | 0.076 | *0.254* | 0.067 | *0.110* | 0.046 |
| | Power | **0.950** | 0.784 | *0.971* | 0.935 | *0.956* | 0.928 |

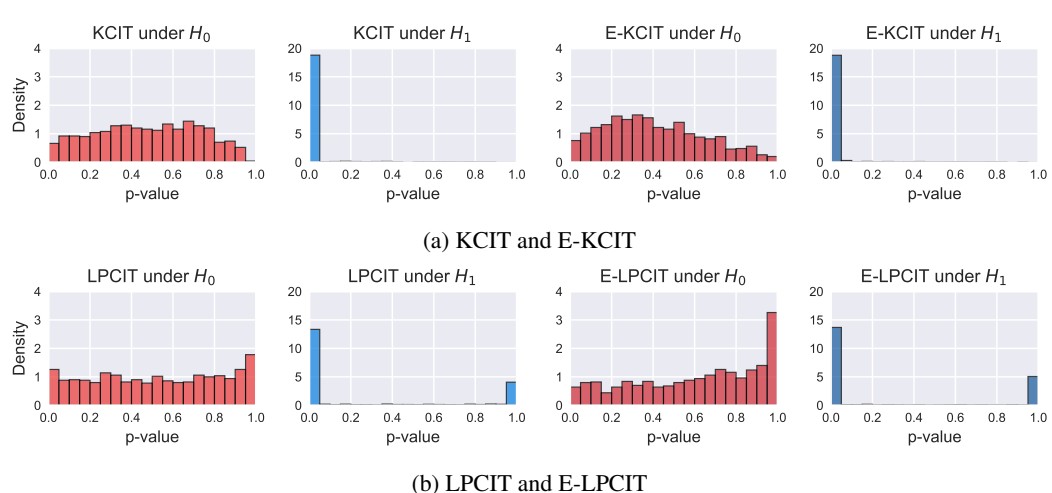

(a) KCIT and E-KCIT

(b) LPCIT and E-LPCIT

Figure 8: Empirical distribution of KCIT, E-KCIT, LPCIT, and E-LPCIT under the null and alternative hypotheses. Results are obtained under the post-nonlinear setup of Section 4.1 with $n = 2000$, $n_k = 400$, and standard $t$-distributed noise (df $= 3$), with a total of 1000 replications.

## F  LIMITATIONS AND FUTURE DIRECTIONS

The primary goal of E-CIT is to provide a general-purpose framework for reducing the computational cost of CIT. Many in-depth discussions that require access to the alternative hypothesis distribution of the base CIT methods fall outside the main scope of this work, but they represent important directions for future research. We discuss several key points below:

**Type I Error Control.** Controlling Type I error is a central problem in hypothesis testing and remains one of the major challenges for CIT. In Theorem 1, we provide theoretical guarantees for Type I error control of E-CIT, in the ideal case where the subtest p-values are perfect. Although the asymptotic validity of specific CIT methods is not the focus of this work, in practice, deviations of subtest p-values from the Uniform$(0, 1)$ may induce shifts in Type I error, which can be amplified by E-CIT. As illustrated in Figure 8, the p-value distributions of the original and ensemble versions of KCIT and LPCIT under the null and alternative hypotheses show that under the null, KCIT p-values are slightly deflated at both ends, while LPCIT p-values are slightly inflated. These deviations lead to right-skew and left-skew in their ensemble versions, which may result in slightly higher or lower Type I errors.

Such deviations are not necessarily detrimental. Indeed, we observe many ensemble versions exhibiting slightly lower Type I error. Nonetheless, Type I error should ideally be maintained at the nominal level in hypothesis testing. Although this is challenging in the context of CIT, it is possible to minimize such deviations within the E-CIT framework in practice. We recommend that the subtest sample size $n_k$ be sufficiently large to ensure that the performance requirements under the alternative hypothesis (Theorem 2) are satisfied, while also guaranteeing sufficiently good asymptotic

$\text{Uniform}(0, 1)$ behavior under the null. Our ablation studies on subset size in Appendix E.6 support this recommendation.

**Super-Uniform p-values from Permutation Tests.** Many CIT methods, particularly those based on the Conditional Randomization Test (CRT), obtain p-values via permutation. These p-values are super-uniform under the null, meaning that for any $a \in [0, 1]$, $P(p \leq a) \leq a$. The discrete, stepwise nature of their CDF can lead to conservative behavior. However, as the number of permutations increases, the super-uniform distribution approaches a strict uniform distribution, and this theoretical convergence justifies the approximation. The validity guarantee in Theorem 1 relies on transforming strictly uniform p-values to a stable distribution via the inverse probability integral transform. Applying the same transform to super-uniform p-values does not yield an exact stable distribution, but with a sufficiently large number of permutations, the resulting distribution closely approximates the ideal transformation. Hence, in practice, super-uniform p-values from permutation tests do not significantly impair the performance of E-CIT.

Additionally, permutation-based p-values may take values exactly equal to 0 or 1, corresponding to extreme rejection or acceptance of the null. Although these values are within the framework's definition, they can create practical computational issues. We suggest adding a small uniform random perturbation to permutation p-values to avoid such issues and to produce a more continuous distribution, closer to strict uniformity.

It should be noted that for permutation-based CIT, the computational savings of E-CIT are limited, as the primary cost arises not from sample complexity but from repeated permutations. Developing methods that specifically reduce the computational burden of the permutation procedure itself represents an important direction for improving the practical scalability of causal discovery algorithms.

**Compatibility of the Ensemble Framework with CIT and Convergence Rates.** Our experiments indicate that E-CIT achieves competitive or even superior test power. The key factor is the comparison of two convergence rates: whether adding more samples directly to a single test or performing multiple subtests and aggregating leads to faster growth in test power. For traditional parametric tests, power grows rapidly with sample size, making separate subtests inefficient. In contrast, for CIT, both the inherent convergence limitations of the base test and the difficulty of satisfying its consistency assumptions in practice suggest that separate subtests may be more efficient. While prior work has investigated the convergence rates of specific CIT methods (Jamshidi et al., 2024; Balabdaoui et al., 2025), establishing theoretical guarantees comparing the two strategies remains a fundamental open problem.

**Hyperparameter Selection and Theoretical Optimality.** As a general framework, E-CIT does not require access to the alternative hypothesis distribution and therefore cannot provide a theoretically optimal combination strategy, particularly with respect to the stability parameter $\alpha$. Currently, our framework primarily offers flexibility that allows adaptation to different CIT methods and data distributions. Empirical choices for $\alpha$ are discussed in Appendix E.1. Future work could explore the framework's full potential by investigating theoretically optimal aggregation strategies for specific CIT methods based on their properties under the alternative hypothesis. While this is a complex problem, we currently recommend using empirical values ($\alpha = 1.75$ or $2$), which, although not theoretically optimal, have proven to be effective in our experiments.

The second core hyperparameter is the number $K$ of subtests (or equivalently, the subset size $n_k$). In practice, it is sufficient to ensure that each subset is large enough to maintain good asymptotic $\text{Uniform}(0, 1)$ behavior under the null and adequate power under the alternative of the subtests, so that Theorems 1 and 2 hold approximately. The choice of $n_k$ can be guided by empirical results from studies of the original CIT methods, balancing statistical performance and computational cost.

**The Curse of Dimensionality.** The curse of dimensionality is a fundamental challenge in CIT. While our experiments on real-world datasets encompass varying dimensions for the conditioning set $Z$, with a dedicated empirical analysis in Appendix E.3, addressing this issue directly is not the primary objective of the E-CIT framework. Instead, E-CIT is designed to be orthogonal to the internal mechanisms of base tests. Indeed, numerous specialized CIT methods (Sen et al., 2017; Bellot & van der Schaar, 2019; Shi et al., 2021; Li et al., 2023a;b) have already been developed to handle the high-dimensional conditioning sets $Z$, and E-CIT can serve as a scalable wrapper for these methods.

**Correlated p-values.** One may consider using resampling methods such as bootstrap to generate additional p-values and improve small-sample test performance. In fact, a similar idea has already been validated in small-sample causal discovery scenarios (Xiang et al., 2024). However, this approach inevitably introduces correlations among p-values. In CIT, the theoretical form of the p-value distribution under the alternative is inherently challenging, making it difficult to model correlations between tests, unlike in the parametric setting studied by (Liu & Xie, 2020; Ling & Rho, 2022). While E-CIT focuses on reducing computational cost in large-sample settings, exploring strategies to exploit overlapping data splits or correlated subtests is an interesting direction for improving CIT performance.

**Distribution Drifts.** The current theoretical guarantees of E-CIT rely on the assumption that the $K$ subtests are independent and identical, resulting in i.i.d. p-values. However, in many real-world scenarios, such as data collected across heterogeneous environments or time-series data, the underlying data-generating mechanisms may experience distribution shifts. In such instances, the theoretical guarantees of our current framework may not strictly hold. Understanding how these drifts impact the aggregated p-value, and developing adaptive subset partitioning or robust aggregation strategies to enhance resilience, represent a highly practical direction for future research.

**Method-specific Enhancements.** E-CIT is designed as a general framework and does not incorporate method-specific optimizations. While we demonstrate its effectiveness across multiple CIT methods, further improvements may be possible by integrating E-CIT more deeply with specific methods. For example, DGCIT (L-folds) (Shi et al., 2021) and NNSCIT (3-folds) (Li et al., 2023a) internally use data splitting. Combining E-CIT principles with these internal schemes may further enhance sample efficiency.

