# OpenReview forum: "Efficient Ensemble Conditional Independence Test Framework for Causal Discovery"
_ICLR.cc/2026/Conference — ICLR 2026 Poster_

### Official Review · Reviewer_7jSe · 2025-10-31

**Soundness:** 3
**Presentation:** 3
**Contribution:** 3
**Rating:** 6
**Confidence:** 3

**Summary:**

The paper is related to causal discovery and proposes a generic Ensemble Conditional Independence Test (E-CIT), which uses a divide-and-aggregate strategy to reduce the typical superlinear numerical complexity to a linear one. The approach is compatible with various existing CIT methods. Several numerical experiments are performed to illustrate the performance of the ensemble approach. Results for different baseline setups look promising. Type I error and statistical power remain competitive while saving computational time.

**Strengths:**

S1 The considered problem is interesting and relevant.
S2 The paper is technically sound and overall well-written.
S3 To obtain linear complexity is advantageous.
S4 The evaluation and comparison against baselines is convincing.

**Weaknesses:**

W1 It remains unclear whether hyper parameters have to be tuned.
W2 Limitations of the paper’s methods could be better discussed.

**Questions:**

Does it make a difference how the data is partitioned into the K subsets?

How to automatically choose the hyperparameter \alpha in general applications? How to choose the number of samples K?

Are the results worse in Table 2 and 3 if alpha is chosen smaller or larger?

The left windows in Figure 2 for the Type I Error do not have a good y-axis scaling. The key differences cannot be seen. Please use a e.g. range between 0.0 and 0.08.

Under which conditions the approach performs not good (e.g. regarding the data partitioning)?

Minor comments:

- line 391,408,410: Why are some numbers italic? What does it mean?

---

> ### Author Response · Authors · 2025-11-19
> **Response to Reviewer 7jSe**
>
> We sincerely thank you for your diligent review and valuable feedback, which greatly improved our manuscript's quality and clarity. Please first refer to our General Response, where we address shared concerns regarding Methodological Novelty and Hyperparameter Settings (your W1, Q2, Q3). Below, we provide detailed responses to your specific comments. If anything remains unclear or if you have further questions, please do not hesitate to let us know.
>
>
> > W2. Limitations of the paper’s methods could be better discussed.
>
> Thank you for raising this critical and constructive point regarding the discussion of our method’s limitations. We fully agree that a transparent discussion is essential for responsible research.
>
> We provide a two-page analysis in **Appendix F Limitations and Future Directions**, covering aspects like Type I Error, super-uniform p-values, hyperparameter optimality, and computational trade-offs with permutation tests.
>
> As you correctly pointed out, these discussions were not sufficiently highlighted in the main text. We have therefore strengthened Section 5 in the revised version to clearly guide readers to these limitations.
>
>
>
> > Q1. Does it make a difference how the data is partitioned into the K subsets?
> >
> > Q5. Under which conditions the approach performs not good (e.g. regarding the data partitioning)?
>
> Thank you for raising this crucial question.
>
> **Partitioning Strategy**: must be random and disjoint
> - Any **non-random or systematic partitioning** (e.g., stratification by Z as in FastKCIT [1]) breaks the "identical" condition, violates our theoretical assumptions, and contradicts the general-purpose goal.
> - **Resampling-based strategies** break independence and violate our theoretical assumptions as well, though they may be promising for small-sample causal discovery. We discuss this direction in Appendix F _Correlated p-values_.
>
>
> **Effect of hyperparameters**: As noted by the reviewer, the partition parameters ($n_k$ or $K$) affect performance:
> - **Theoretically**:
>     - Too small $n_k$ leads to unstable subtest p-values, which can harm Type I error control.
>     - Too small $n_k$ may also cause subtests to violate the conditions of Theorem 2.
>     - But smaller $n_k$ increases $K$, which may accelerate convergence under the conditions of Theorem 2, this benefit requires the assumptions to hold.
> - **Empirically**:
> 	- Appendix E.6 (Figure 7) shows that very small $n_k$ harms Type I error control, while the effect on power is mixed (sometimes better, sometimes worse), which is consistent with the theoretical predictions.
>
> Meanwhile, poor choices of $\alpha$ (failing Theorem 2) can reduce performance, as observed in Appendix E.1.
>
> > Q3. Are the results worse in Table 2 and 3 if alpha is chosen smaller or larger?
>
> Not necessarily; performance could improve or degrade depending on the specific setting. As discussed in Lines 268–281, the theoretically optimal $\alpha$ is context-dependent (varying by base CIT and data distribution). While our recommended defaults are empirically robust across diverse scenarios, tuning $\alpha$ could potentially yield better results if it moves closer to the specific optimum. However, selecting an $\alpha$ that violates the consistency conditions of Theorem 2 will degrade performance, as demonstrated in Appendix E.1.
>
>
>
> > Q4. The left windows in Figure 2 for the Type I Error do not have a good y-axis scaling. The key differences cannot be seen. Please use a e.g. range between 0.0 and 0.08.
> >
> > Q6. Minor comments: - line 391,408,410: Why are some numbers italic? What does it mean?
>
> We thank the reviewer for the careful attention to detail.
> - Following your suggestion, the y-axis scaling of Figure 2 has been updated in the revised version.
> - Regarding italic numbers: the meaning was explained in lines 374–376 of the original version ("bold indicates significant improvement of ensemble over base; bold italics indicate the opposite"). We agree the explanation was not sufficiently visible, and have now repeated this clarification directly in each table caption.
>
> ---
>
> [1] Oliver Schacht and Biwei Huang. A fast kernel-based conditional independence test with application to causal discovery. arXiv preprint arXiv:2505.11085, 2025.

---

> ### Author Response · Authors · 2025-11-27
> **Looking forward to discussion**
>
> Dear Reviewer 7jSe,
>
> As the discussion period is coming to a close, we wanted to check back to see whether you have any remaining questions. We are truly grateful for your detailed and constructive suggestions, based on which we have further polished the paper. We really appreciate your time engaged in the review process and look forward to your reply.
>
> Best regards, Authors

---

### Official Review · Reviewer_cUG3 · 2025-10-31

**Soundness:** 3
**Presentation:** 3
**Contribution:** 2
**Rating:** 4
**Confidence:** 3

**Summary:**

This paper proposes E-CIT, a framework for scalable conditional independence testing (CIT) that divides data into subsets, applies any base CIT independently, and aggregates p-values through a stable-distribution–based combination. The goal is to reduce computational cost while preserving Type I error control and statistical stability. The paper provides theoretical proofs of validity and convergence, and empirical results on synthetic and real datasets showing comparable accuracy to full-sample CITs but much better efficiency.

**Strengths:**

1. The paper addresses a practical bottleneck in causal discovery with a general framework that can wrap around existing tests such as without retraining.

2. The use of stable-distribution aggregation is mathematically sound and provides more stable inference than classical methods, with clear theoretical foundation.

3. Experiments show that E-CIT achieves similar Type I and power performance as full CIT while reducing computation time by up to an order of magnitude on large datasets.

**Weaknesses:**

1. The proposed framework is mainly an adaptation of existing ensemble ideas; it does not introduce a fundamentally new conditional independence test or inference principle.

2. Although the theoretical proofs are solid, they mostly re-establish known results from p-value combination theory in the CIT context, making the methodological novelty limited.

3. The evaluation focuses mostly on synthetic data and standard causal discovery tasks; there is little evidence on whether E-CIT maintains robustness on high-dimensional or real-world noisy domains.

**Questions:**

Since E-CIT combines subset-level p-values, how sensitive is it to the number of partitions. does too few or too many subsets affect power consistency?

The theoretical analysis assumes independence between subsets; in real data where partitions can still share dependence structure, how much does this violate the validity guarantees?

Could the authors compare their stable-distribution aggregation directly against the more recent Cauchy Combination Test (Liu & Xie 2020) to clarify relative advantages?

---

> ### Author Response · Authors · 2025-11-19
> **Response to Reviewer cUG3**
>
> We sincerely appreciate your time and effort in reviewing our paper. Please first refer to our General Response, where we address shared concerns regarding Methodological Novelty (your W1, W2, Q3) and Hyperparameter Settings (your Q1). Below, we provide detailed responses to your specific comments. If anything remains unclear or if you have further questions, please do not hesitate to let us know.
>
> > W3. The evaluation focuses mostly on synthetic data and standard causal discovery tasks; there is little evidence on whether E-CIT maintains robustness on high-dimensional or real-world noisy domains.
>
> We agree on the importance of robustness evaluation. We provide several forms of evidence supporting the robustness of E-CIT:
> - On real-world data, we demonstrate advantages on the widely used Flow Cytometry Dataset, and we added a smaller-sample study in Appendix E.4 as suggested by Reviewer CASq.
> - Although large conditioning sets and large sample sizes (n) present orthogonal challenges, we also evaluate related experiments in Appendix E.3, showing robustness of the method.
>
> > Q1. Since E-CIT combines subset-level p-values, how sensitive is it to the number of partitions. does too few or too many subsets affect power consistency?
>
> Yes, the partition size ($n_k$ or $K$) affects performance:
> - **Theoretically**:
>     - Too small $n_k$ leads to unstable subtest p-values, which can harm Type I error control.
>     - Too small $n_k$ may also cause subtests to violate the conditions of Theorem 2.
>     - But smaller $n_k$ increases $K$, which may accelerate convergence under the conditions of Theorem 2, this benefit requires the assumptions to hold.
> - **Empirically**:
> 	- Appendix E.6 (Figure 7) shows that very small $n_k$ harms Type I error control, while the effect on power is mixed (sometimes better, sometimes worse), which is consistent with the theoretical predictions.
>
> Crucially, sensitivity to $n_k$ is mild once it exceeds a basic threshold. Figure 7 demonstrates consistent performance across a broad range of configurations. Furthermore, by following our hyperparameter guideline (General Response) selecting an appropriate partition size is straightforward in practice.
>
> > Q2. The theoretical analysis assumes independence between subsets; in real data where partitions can still share dependence structure, how much does this violate the validity guarantees?
>
> This is an excellent statistical question. Under the standard **I.I.D. assumption**, random disjoint partitions are _statistically independent_; thus our theoretical assumptions are satisfied.
>
> The subtle distinction is:
> - **Finite population perspective:** If the dataset $D$ is treated as a fixed population and subsets are drawn _without replacement_, then $D_1$ and $D_2$ are indeed correlated.
> - **Statistical inference perspective (ours):** As in most machine learning and statistical analysis, we consider the data as i.i.d. samples from an underlying distribution. Under this assumption, randomly partitioned subsets $D_1$ and $D_2$ are _statistically independent_, and therefore their p-values satisfy Definition 1.
>
> We also note that handling _correlated p-values_ in ensemble testing is an interesting future direction with potential relevance to small-sample causal discovery. We discuss this in Appendix F _Correlated p-values_.

---

> ### Author Response · Authors · 2025-11-27
> **Looking forward to discussion**
>
> Dear Reviewer cUG3,
>
> As the discussion period is coming to a close, we wanted to check back to see whether you have any remaining questions. We have specifically clarified the **fundamental distinction** between ours and existing methods, and highlighted the robustness of our framework. We really appreciate your time engaged in the review process and look forward to your reply.
>
> Best regards, Authors

---

### Official Review · Reviewer_nRmx · 2025-11-01

**Soundness:** 3
**Presentation:** 3
**Contribution:** 2
**Rating:** 4
**Confidence:** 3

**Summary:**

This paper introduces E-CIT, a divide-and-aggregate framework to solve the high computational cost of conditional independence (ci) tests. E-CIT partitions the data into subsets, applies base CI tests to each subset, and combines the resulting p-values. Its core contribution is a p-value combination method based on stable distributions, which provides theoretical consistency. This framework reduces the complexity of base CITs to linear in sample size (when subset size is fixed) and demonstrates competitive or superior power, especially in heavy-tailed scenarios.

**Strengths:**

1. The paper is well-structured, and the proposed framework is also intuitive.
2. The framework can be combined with different CI methods.
3. Table 1 is clear and useful for unfamiliar readers.

**Weaknesses:**

1. It is intuitive to divide the sample set into several subsets to reduce computational burden. However, I find it difficult to understand why this simultaneously enhances statistical power in experiments. The theoretical justification throughout the paper does not focus on or support this point. Consequently, we are still unclear under which conditions the E-CIT framework outperforms single-test execution.
2. The paper's core theoretical guarantees depend on the p-values from the subtests being well-defined and effective. To ensure the subtest p-values are accurate, $n_k$ should be sufficiently large and tend to infinity. On the other hand, the paper claims that the framework reduces time complexity to $O(n)$ when $n_k$ is treated as a fixed constant, which raises a contradiction. $n_k$ cannot be simultaneously treated as a small, fixed constant to justify the $O(n)$ complexity claim, and as an asymptotically large number to justify the theoretical validity and consistency of the theorems.
3. The paper states that the median heuristic is used for all methods to ensure a fair comparison. However, bandwidth is sample-size dependent. To isolate the effect of the ensemble strategy, I suggest the authors attempt to find the optimal bandwidth for each test in its respective setting (e.g., via bandwidth optimization [1]).
4. The framework does not provide a principled or automated method to select $n_k$.
5. The authors claim in the contribution that the p-value combination method proposed is novel. It may be slightly overstated. The method shares a related technical foundation with existing works on p-value combination (e.g., Ling & Rho (2022), Liu & Xie (2020)), which are already mentioned in the paper's related work.

References:
[1]  W. Wang et al, Practical Kernel Selection for Kernel-based Conditional Independence Test. NeurIPS 2025.

**Questions:**

1. Is $n_k=400$ still a good choice with a larger sample size?
2. Please elaborate more details on the differences and the relation with Stouffer et al. (1949).

---

> ### Author Response · Authors · 2025-11-19
> **Response to Reviewer nRmx**
>
> We sincerely appreciate your time and effort in reviewing our paper. Please first refer to our General Response, where we address shared concerns regarding Methodological Novelty (your W5, Q2) and Hyperparameter Settings (your W4, Q1). Below, we provide detailed responses to your specific comments. If anything remains unclear or if you have further questions, please do not hesitate to let us know.
>
> > W1. It is intuitive to divide the sample set into several subsets to reduce computational burden. However, I find it difficult to understand why this simultaneously enhances statistical power in experiments......under which conditions the E-CIT framework outperforms single-test execution.
>
> We thank the reviewer for raising this deep and interesting point. We fully understand the intuition that partitioning may appear to reduce power—our observation may indeed be the first to document the opposite phenomenon.
>
> E-CIT tends to gain advantages in **more challenging testing scenarios**. As discussed in Appendix C and F _Compatibility of the Ensemble Framework with CIT and Convergence Rates_, CIT is inherently difficult and the power can converge slowly with sample size. For example, in Figure 2(a) the power of KCIT saturates after $n > 1600$. Thus, the marginal benefit of adding more samples to a single test may be smaller than the benefit of forming an _additional_ reasonably effective subtest and aggregating them (This phenomenon is consistent with the classical ensemble-learning intuition, where averaging moderately informative weak learners yields stronger results than a single strong learner).
>
> This interpretation is further **corroborated by our experimental observations:** E-CIT performs strongly in difficult settings (Fig. 2(a), 2(b)) yet offers no significant advantage in simpler scenarios (Fig. 2(c) or the Gaussian-noise case detailed in our response to Reviewer CASq, Q5).
>
> > W2. The paper's core theoretical guarantees depend on the p-values from the subtests being well-defined and effective......$n_k$ cannot be simultaneously treated as a small, fixed constant to justify the $O(n)$ complexity claim, and as an asymptotically large number to justify the theoretical validity and consistency of the theorems.
>
> Thank you for giving us the opportunity to clarify this central design trade-off. We believe there is **no contradiction**: there exists a practical and effective range of $n_k$ that is _small_ relative to total cost but _large enough_ for good asymptotic behavior.
>
> Although theoretical validity of p-values in modern CITs is asymptotic, this does **not** require literally letting $n_k \to \infty$. This mirrors the application of the Central Limit Theorem: while theoretically is asymptotic, in practice, a sample size of 30 can already make a sum nearly indistinguishable from Gaussian. Similarly, many CIT methods empirically achieve well-behaved p-values at moderate sample sizes (e.g., $n_k$ = 400 for KCIT, as seen in Figure 8). Thus, $n_k$ does not need to be both large and small at the same time. Rather, we choose $n_k$ to be **large enough** for subtest validity but **small enough** to ensure linear-time scaling.
>
> Additionally, Theorem 2 concerns **consistency under** $H_1$ and requires only that subtests be reasonably effective, which does not rely on $n_k \to \infty$.
>
> We acknowledge that a theoretical–practical gap may still exist, which is why we examined Type I error in all experiments and provided detailed analysis of $n_k$ in Appendix E.6 and F.
>
> > W3. The paper states that the median heuristic is used for all methods to ensure a fair comparison. However, bandwidth is sample-size dependent. To isolate the effect of the ensemble strategy, I suggest the authors attempt to find the optimal bandwidth for each test in its respective setting (e.g., via bandwidth optimization [1]).
>
> We thank the reviewer for the thoughtful suggestion. We fully agree that bandwidth selection is a critical factor. However, we employed the median heuristic specifically to ensure **the fairest and most controlled comparison** within the ensemble setting.
>
> Since the median heuristic adapts dynamically to the pairwise distances within each dataset, it provides a **standardized and consistent** bandwidth baseline for all methods. This approach effectively isolates the impact of the ensemble strategy, avoids potential confounding factors that could be introduced by complex optimization routines. While bandwidth optimization is theoretically appealing, it introduces algorithmic variations unrelated to the ensemble mechanism itself.
>
> Nevertheless, we acknowledge that bandwidth optimization [1] is a significant area of research in kernel-based CITs. Accordingly, we have incorporated a discussion and relevant citations on this topic in the revised manuscript.
>
> ---
>
> [1] W. Wang et al, Practical Kernel Selection for Kernel-based Conditional Independence Test. NeurIPS 2025.

---

> ### Author Response · Authors · 2025-11-27
> **Looking forward to discussion**
>
> Dear Reviewer nRmx,
>
> As the discussion period is coming to a close, we wanted to check back to see whether you have any remaining questions. We have specifically clarified the **fundamental distinction** between ours and existing methods, and addressed your insightful questions. We really appreciate your time engaged in the review process and look forward to your reply.
>
> Best regards, Authors

---

### Official Review · Reviewer_CASq · 2025-11-11

**Soundness:** 3
**Presentation:** 3
**Contribution:** 2
**Rating:** 4
**Confidence:** 3

**Summary:**

This paper proposes an efficient ensemble independence testing approach that reduces computational cost by first partitioning the data into subsets, then applying a base conditional independence test (CIT) on each subset, and finally aggregating the resulting p-values. This strategy reduces the complexity of a base CIT to be linear in the sample size. Empirical results show that the proposed method improves efficiency while maintaining comparable accuracy.

**Strengths:**

-	The paper presents a strong motivation and a practical approach for improving the efficiency of conditional independence tests, and consequently, constraint-based causal discovery.
-	The paper is well-written and clearly presented.

**Weaknesses:**

-	This paper provides limited evaluation of generality; see Questions 5, 6, and 7 for details.
-	The paper makes strong claims of optimality but lacks sensitivity analysis for violations of the theorem’s assumptions, some of which appear overly strong.

**Questions:**

1.	Could the authors provide references to support the claim that ensemble learning can address high computational complexity (lines 131–134)?
2.	Is this the first work to propose an ensemble-based conditional independence testing approach?
3.	What happens if the assumptions in Definition 1 are violated? Does the derived ensemble test still hold under such conditions?
4.	What are the implications if the unbiasedness assumption in Theorem 1 is violated? For instance, subtests may be biased in practice if the selected data subsets are not representative. While I understand there is a theoretical–practical gap, it would be helpful to analyze how sensitive the proposed algorithm is to such violations, given that the paper emphasizes its asymptotic optimal properties.
5.	Why do the authors assume nonlinear causal functions? It would be informative to also report results for linear causal functions. Additionally, can the proposed algorithm be applied to data with Gaussian noise? If so, does it still demonstrate advantages in accuracy and efficiency compared to state-of-the-art methods?
6.	The experiments on real data are not entirely convincing. The paper uses synthetically generated interventional data rather than directly collected real-world data. A more rigorous and widely accepted evaluation strategy would rely solely on the 783 observational samples. Otherwise, it is difficult to substantiate the claim that the proposed method performs well on real-world data.
7.	The sample size may not be the dominant factor affecting causal discovery efficiency. The primary challenge for constraint-based approaches lies in the accuracy and scalability of conditional independence tests when the conditioning set is high-dimensional. How does the proposed approach perform under such settings?

---

> ### Author Response · Authors · 2025-11-19
> **Response to Reviewer CASq (Part 1/2)**
>
> We sincerely appreciate your time and effort in reviewing our paper. Please first refer to our General Response, where we address shared concerns regarding Methodological Novelty and Hyperparameter Settings. Below, we provide detailed responses to your specific comments. If anything remains unclear or if you have further questions, please do not hesitate to let us know.
>
> > Q1. Could the authors provide references to support the claim that ensemble learning can address high computational complexity (lines 131–134)?
>
> Certainly. More precisely, E-CIT leverages a **divide-and-conquer principle** to control computational complexity, while the "ensemble" aspect refers to aggregating results to recover statistical power.
>
> Divide-and-conquer is a foundational paradigm for handling computational bottlenecks:
>
> 1. In classical algorithms, it underlies many efficient methods (for example, the linear-time selection algorithms [1]).
> 2. In Big Data Analysis, it is a common strategy for mitigating computational bottlenecks, with many survey papers discussing its role [2].
> 3. In CIT specifically, the contemporaneous FastKCIT [3] employs internal partitioning via Gaussian Mixture Models to achieve fast approximate calculations for KCIT (lines 79–80).
>
>
> > W2. The paper makes strong claims of optimality but lacks sensitivity analysis for violations of the theorem’s assumptions, some of which appear overly strong.
> >
> > Q3. What happens if the assumptions in Definition 1 are violated? Does the derived ensemble test still hold under such conditions?
> >
> > Q4. What are the implications if the unbiasedness assumption in Theorem 1 is violated? For instance, subtests may be biased in practice if the selected data subsets are not representative. While I understand there is a theoretical–practical gap, it would be helpful to analyze how sensitive the proposed algorithm is to such violations, given that the paper emphasizes its asymptotic optimal properties.
>
>
> We thank the reviewer for the careful reading of our theoretical analysis, these questions significantly improve the clarity of the manuscript. We clarify the intended scope and implications of our theorems below.
>
> 1. **Violation of Definition 1(2)**: If the chosen distribution $S$ is not stable (violating Definition 1), the additive closure property (Proposition 1) fails. Consequently, the resulting $p_e$​ would no longer be a valid p-value under the null hypothesis. In this case, the definition of the ensemble test would no longer be meaningful.
> 2. **Violation of Theorem 1 assumptions**: The "unbiasedness" in Theorem 1 refers specifically to **test unbiasedness**, not parameter-estimation unbiasedness (e.g., bias arising from unrepresentative samples). In hypothesis testing, unbiasedness simply implies that the test performs better than **random guessing**, a property satisfied by essentially all meaningful hypothesis tests. Validity means the subtest is a valid hypothesis test; admissibility indicates it is not dominated by another test. Therefore, Theorem 1 serves as a primarily guarantee that our ensemble procedure does not lose the fundamental properties of the underlying subtests. These assumptions are standard and rarely violated in practice.
> 3. **Theorem 2 and sensitivity analysis**: We agree with the reviewer that bridging the theoretical–practical gap is crucial. Theorem 2 is the key practical theorem: its first two conditions require subtests to be "reasonably effective", indirectly constraining $n_k$; the third condition constrains $\alpha$. Accordingly, we conducted sensitivity experiments for both $n_k$ and $\alpha$ (Appendix E.6 and E.1), and provided detailed discussion (lines 242–267; Appendix D; Appendix F). In particular, Figure 8 shows that the empirical distribution ($n_k = 400$) under $H_1$ satisfies the first two conditions of Theorem 2.
> 4. **Clarification on "Optimality":** We do **not** claim to have proposed a universally optimal test; indeed, it is established in CIT literature that no single method is uniformly effective. Our reference to a "uniformly most powerful test" (Line 269) illustrates the **flexibility** of the stable distribution family. According to the Neyman-Pearson lemma, the optimal combination rule changes depending on the specific CIT method and data distribution. Our aggregation method provides flexibility to approximate optimal forms in different settings

---

> ### Author Response · Authors · 2025-11-19
> **Response to Reviewer CASq (Part 2/2)**
>
> > W1. This paper provides limited evaluation of generality; see Questions 5, 6, and 7 for details.
> >
> > Q5. Why do the authors assume nonlinear causal functions? It would be informative to also report results for linear causal functions. Additionally, can the proposed algorithm be applied to data with Gaussian noise? If so, does it still demonstrate advantages in accuracy and efficiency compared to state-of-the-art methods?
>
> We thank the reviewer for carefully examining our experimental design.
>
> We focus on nonlinear / non-Gaussian settings because **these represent the fundamental challenges that modern CIT aims to solve**, while linear or Gaussian settings are already well handled by classical methods. The post-nonlinear model is also standard in many CIT studies.
>
> As suggested, we additionally evaluated the same settings in Figure 2 but with Gaussian noise (Type I error behaves well and is omitted):
>
> | Power vs Sample Size | 800   | 1600  |
> | -------------------- | ----- | ----- |
> | KCIT                 | 0.949 | 0.974 |
> | RCIT                 | 0.958 | 0.984 |
> | FastKCIT             | 0.944 | 0.957 |
> | E-KCIT ($\alpha=2$)  | 0.938 | 0.964 |
>
> We also evaluated strictly linear functions:
>
> | Power vs Sample Size | 800   | 1600  |
> | -------------------- | ----- | ----- |
> | KCIT                 | 1.000 | 1.000 |
> | RCIT                 | 0.996 | 0.999 |
> | FastKCIT             | 1.000 | 1.000 |
> | E-KCIT ($\alpha=2$)  | 1.000 | 1.000 |
>
> The power of all methods saturates (reaching 1.0) even at moderate sample sizes, making meaningful performance comparison impossible. This is why we, like other modern CIT studies, focus on the more discriminative non-linear settings.
>
>
>
> > Q6. The experiments on real data are not entirely convincing. The paper uses synthetically generated interventional data rather than directly collected real-world data. A more rigorous and widely accepted evaluation strategy would rely solely on the 783 observational samples. Otherwise, it is difficult to substantiate the claim that the proposed method performs well on real-world data.
>
> We respectfully clarify a misunderstanding regarding the nature of the data: the 1,755 samples used are **real biological data**, **not synthetic data**. These samples come from Sachs et al. [4]. The confusion likely stems from the term "stimulation" in the original dataset description, which refers to **biological reagents** (e.g., CD3, CD28) used to stimulate cells, not computer "simulations".
>
> | Stimulation:        | File             |
> | ------------------- | ---------------- |
> | 1.CD3, CD28         | cd3cd28.xls      |
> | 2.CD3, CD28, ICAM-2 | cd3cd28icam2.xls |
> | ......              | .......          |
>
> Following recent literature [5], we utilize the datasets corresponding to the first two experimental conditions. In these settings, the stimulations act on unobserved upstream variables, which serves as a source of exogenous variation **without altering the underlying causal mechanisms** among the observed proteins. Therefore, the causal structure remains invariant, and the data can be effectively treated as **observational** for the purpose of CIT. We have revised Appendix E.4 to remove any ambiguity.
>
> We also appreciate the suggestion regarding the 783 sample (likely the 853-sample `cd3cd28` file). We have added these experiments to the revised Appendix E.4, which can align with other works using this specific dataset [6]. Although small-sample scenarios are not the primary target of E-CIT, the method still shows advantages.
>
>
> ---
>
> [1] Blum, M., Floyd, R. W., Pratt, V. R., Rivest, R. L., & Tarjan, R. E. Time bounds for selection. J. Comput. Syst. Sci., 7(4), 448-461, 1973.
>
> [2] Chen, X., Cheng, J. Q., & Xie, M. G. Divide-and-conquer methods for big data analysis. arXiv preprint arXiv:2102.10771, 2021.
>
> [3] Oliver Schacht and Biwei Huang. A fast kernel-based conditional independence test with application to causal discovery. arXiv preprint arXiv:2505.11085, 2025.
>
> [4] Karen Sachs, Omar Perez, Dana Pe’er, Douglas A Lauffenburger, and Garry P Nolan. Causal proteinsignaling networks derived from multiparameter single-cell data. Science, 308(5721):523–529, 2005.
>
> [5] Shuai Li, Ziqi Chen, Hongtu Zhu, Christina Dan Wang, and Wang Wen. Nearest-neighbor sampling based conditional independence testing. In Proceedings of the AAAI Conference on Artificial Intelligence, volume 37, pp. 8631–8639, 2023.
>
> [6] Sudipto Mukherjee, Himanshu Asnani, and Sreeram Kannan. Ccmi: Classifier based conditional mutual information estimation. In Uncertainty in artificial intelligence, pp. 1083–1093. PMLR, 2020.

---

> ### Author Response · Authors · 2025-11-27
> **Looking forward to discussion**
>
> Dear Reviewer CASq,
>
> As the discussion period is coming to a close, we wanted to check back to see whether you have any remaining questions. We have clarified the experimental settings and real data usage, and discussed the sensitivity to assumption violations. We really appreciate your time engaged in the review process and look forward to your reply.
>
> Best regards, Authors

---

### Author Response · Authors · 2025-11-19
**General Response to Reviewers (Part 1/2, Methodological Novelty)**

**Methodological Novelty:**

> Reviewer CASq:
>
> - Q2. Is this the first work to propose an ensemble-based conditional independence testing approach?
> - Q7. The sample size may not be the dominant factor affecting causal discovery efficiency...
>
> Reviewer nRmx:
>
> - W5. The authors claim in the contribution that the p-value combination method proposed is novel. It may be slightly overstated. The method shares a related technical foundation with existing works on p-value combination (e.g., Ling & Rho (2022), Liu & Xie (2020)), which are already mentioned in the paper's related work.
>
> - Q2. Please elaborate more details on the differences and the relation with Stouffer et al. (1949).
>
> Reviewer cUG3:
>
> - W1. The proposed framework is mainly an adaptation of existing ensemble ideas; it does not introduce a fundamentally new conditional independence test or inference principle.
>
> - W2. Although the theoretical proofs are solid, they mostly re-establish known results from p-value combination theory in the CIT context, making the methodological novelty limited.
>
> - Q3. Could the authors compare their stable-distribution aggregation directly against the more recent Cauchy Combination Test (Liu & Xie 2020) to clarify relative advantages?


To directly address the reviewers’ core concerns regarding novelty:
- E-CIT is, to the best of our knowledge, the first general ensemble-based framework for conditional independence testing.
- Our p-value combination rule is new both in its mathematical form and in its theoretical setting.
- Our theoretical results do not re-establish prior combination-test theory; they address a more general, subtest-agnostic regime not covered in existing works.

Below we provide clarifications in three parts.

### 1. Novelty and generality of the E-CIT framework

-  To the best of our knowledge, E-CIT is the **first ensemble-based** and the **first general-purpose divide–aggregate framework** for CIT, motivated by the computational barrier caused by the sample-size dimension in modern CITs.
- E-CIT is also, to the best of our knowledge, the **first general framework that directly addresses the computational intractability arising from large $n$** in CIT and causal discovery. Since **no** single CIT is uniformly effective across dependence structures [1], a plug-and-play ensemble framework may be more practically valuable than a single fast CIT.

### 2. Novelty of the p-value combination

- **Our test statistic and aggregation formula are new.** Our ensemble statistic is $T_e \propto \sum F_S^{-1}(p_k),$ an extension of the Stouffer method [2] (when $S$ is standard normal) based on stable distributions. In contrast, Ling & Rho [3] extend Liu & Xie [4] and Lipták [5], using $T \propto \sum F_{S'}^{-1}(1-p_k),$ which includes the Cauchy and Gaussian forms as special cases. When $S$ and $S'$ are the same symmetric stable distribution, these two forms coincide statistically. Accordingly, the Cauchy Combination Test [4] corresponds to our framework with $\alpha = 1, \beta = 0$, which is also included as a hyperparameter setting in Appendix E.1.

- **Our theoretical analysis is fundamentally different from prior work.** As stated in lines 103–107 and 197–202, our results make no parametric assumptions on the form of the subtests. In contrast, Ling & Rho [3] and Liu & Xie [4] rely on normality assumptions for their subtest statistics. Our theoretical setting is more general, and the resulting theory and aggregation form are therefore different.

- **Our motivation and target application differ from all prior p-value combination works.** Existing methods are designed for _meta-analysis_, typically combining p-values across repeated experiments such as biological assays, where test statistics are often Gaussian. Our setting is entirely different: we aggregate, complex CITs whose statistics do **not** have simple parametric forms. This distinct purpose necessitates a fundamentally different theoretical foundation and combination form.

### 3. Contribution Focus: Addressing the Sample-Size Bottleneck

We fully agree that high-dimensional conditioning sets are a central challenge in CIT and causal discovery, and we include related experiments in Appendix E.3. However, the sample-size dimension (n) introduces an **orthogonal** computational bottleneck that is largely overlooked despite being crucial for real-world causal discovery pipelines. E-CIT fills this gap by providing a general, scalable solution. Advances in both dimensions (sample size and conditioning-set dimensionality) are complementary and necessary for practical scalability.

---

> ### Author Response · Authors · 2025-11-19
> **General Response to Reviewers (Part 2/2, Hyperparameter Settings)**
>
> **Hyperparameter Settings:**
>
> > Reviewer nRmx:
> >
> > - W4. The framework does not provide a principled or automated method to select $n_k$.
> >
> > - Q1. Is $n_k=400$ still a good choice with a larger sample size?
> >
> > Reviewer cUG3:
> >
> > - Q1. Since E-CIT combines subset-level p-values, how sensitive is it to the number of partitions. does too few or too many subsets affect power consistency?
> >
> > Reviewer 7jSe:
> >
> > - W1. It remains unclear whether hyper parameters have to be tuned.
> >
> > - Q2. How to automatically choose the hyperparameter \alpha in general applications? How to choose the number of samples K?
> >
> > - Q3. Are the results worse in Table 2 and 3 if alpha is chosen smaller or larger?
> >
> > - Q5. Under which conditions the approach performs not good (e.g. regarding the data partitioning)?
>
> We thank the reviewers for their insightful questions regarding hyperparameters. We realize that our discussion on this topic was previously dispersed across appendices. We provide a consolidated summary below and have revised the main text (Section 5) to explicitly clarify these settings.
>
> Summary of guideline (Appendix F, _Hyperparameter Selection and Theoretical Optimality_):
> - $\alpha$: We recommend a default of $1.75$ or $2$.
> - $n_k$($K$): We recommend referencing the reasonable sample size reported in the original studies of the base CIT (e.g., $n_k=400$ for KCIT [6]).
>
> Regarding $\alpha$:
> 1. **Principle**: We recommend setting $1.75$ or $2$. Both values are stable and strong defaults in all experiments.
> 2. **Design Philosophy**: The flexibility of $\alpha$ is intentionally designed as a _theoretical advantage_, not a tuning burden. As lines 268–281 describe, an optimal $\alpha$ may exist under a given CIT or environment. Our framework allows such flexibility for future research, but for current applications the default values are sufficiently robust.
> 3. **Sensitivity**: Performance variations are context-dependent. While our defaults are empirically robust, tuning $\alpha$ towards the specific optimum could theoretically improve results. However, selecting an $\alpha$ that violates the conditions of Theorem 2 will degrade performance, as demonstrated in Appendix E.1.
>
>
> Regarding $n_k$:
> 1. **Principle**: $n_k$ must be sufficient to ensure the subtest's p-values approach uniformity under the null hypothesis and satisfy the basic effectiveness conditions of Theorem 2. It can directly adopt empirical guidelines from the original base CIT literature.
> 2. **Why $n_k​=400$**: We adopted this setting following the KCIT original paper [6], which shows that KCIT’s p-values are already well-behaved at this sample size (as in our Figure 8(a)), while remaining computationally light.
> 3. **Sensitivity**: Theoretically, $n_k$​ balances ensemble convergence against subtest validity (please refer to our detailed responses to Reviewers cUG3 (Q1) and 7jSe (Q5)). Empirically (Appendix E.6), while excessively small $n_k$​ degrades Type I error control, sensitivity becomes mild once a basic threshold is met. Figure 7 confirms that our straightforward practical selection falls comfortably within this robust region.
> 4. **Large Sample Regime:** When the total sample size is extremely large (resulting in a very high number of partitions $K$), it is beneficial to moderately increase ​$n_k$ (e.g., to 600 or 800). This prevents the accumulation of minute asymptotic deviations from the subtests, ensuring robust Type I error control (see Appendix F _Type I Error Control_ and _Hyperparameter Selection and Theoretical Optimality_).
>
> ---
>
> [1] Rajen Dinesh Shah and J. Peters. The hardness of conditional independence testing and the generalised covariance measure. The Annals of Statistics, 2018.
>
> [2] Samuel A. Stouffer, Edward A. Suchman, Leland C. DeVinney, Shirley A. Star, and Robin M. Williams. The American Soldier: Adjustment During Army Life. Princeton University Press, Princeton, 1949.
>
> [3] Xing Ling and Yeonwoo Rho. Stable combination tests. Statistica Sinica, 32:641–644, 2022.
>
> [4] Yaowu Liu and Jun Xie. Cauchy combination test: a powerful test with analytic p-value calculation under arbitrary dependency structures. Journal of the American Statistical Association, 115(529): 393–402, 2020.
>
> [5] Tamás Lipták. On the combination of independent tests = függelten mozgó szintes próbák összevont értékeléséről. *A Magyar Tudományos Akadémia Matematikai Kutató Intézetének Közleményei*, 3(3-4):171–197, 1958.
>
> [6] Kun Zhang, Jonas Peters, Dominik Janzing, and Bernhard Scholkopf. Kernel-based conditional independence test and application in causal discovery. In UAI 2011, Proceedings of the TwentySeventh Conference on Uncertainty in Artificial Intelligence.

---

### Author Response · Authors · 2025-12-02
**Rebuttal Summary for AC Consideration**

Dear Area Chair,

We sincerely appreciate your dedication to the ICLR community during these unusual circumstances. To facilitate your decision-making process, we have prepared a comprehensive and structured summary of our rebuttal **(Note: All cited Appendices and Theorems were already present in the initial submission)**:

**1. Novelty & Significance (General Response Part 1 -> Reviewer CASq Q2,Q7; nRmx W5,Q2; cUG3 W1,W2,Q3)**

- **Pioneering Contribution:** We clarified that E-CIT addresses a critical yet overlooked computational bottleneck regarding sample size in causal discovery. It provides the **first generic solution** (to our knowledge) and introduces the ensemble paradigm to CIT for the first time.

- **Methodological Innovation:** We clarified that our p-value combination method is a **novel generalization** of traditional approaches, supported by a **new non-parametric** theoretical analysis that **eliminates the reliance on normality assumptions** required by prior works. This ensures E-CIT is flexibly compatible with various modern CIT methods, distinguishing it from traditional simple test meta-analyses.


**2. Hyperparameters (General Response Part 2 -> Reviewer nRmx W4,Q1; cUG3 Q1; 7jSe W1,Q2,Q3,Q5)**

- **Clear Guidelines:** We highlighted the discussion and recommended choices in **Appendix F** (_Hyperparameter Selection and Theoretical Optimality_) and strengthened the main text with direct practical guidelines to enhance visibility.

- **Sensitivity:** We clarified the impact of hyperparameter values, demonstrating that they are consistent with our theoretical analysis (Theorem 2, Appendix F) and supported by the sensitivity experiments in **Appendices E.1 and E.6**.


**3. Theoretical Deep Dive**

- **Theoretical–Practical Gap (Reviewer CASq W2, Q3, Q4):** We provided a detailed clarification of the theoretical implications, highlighting both the **analytical support** (Appendices D, E.1, E.6) and the **empirical satisfaction** (Appendix F, Figure 8) of these theoretical conditions.

- **Suitability of Ensembling (Reviewer nRmx W1):** Summarizing **Appendices C and F** (_Compatibility of the Ensemble Framework with CIT and Convergence Rates_), we explained why the challenges of CIT are suited for an ensemble approach.

- **No Contradiction (Reviewer nRmx W2):** We clarified that $n_k$ serves as a constant sufficient for practical validity (akin to finite samples in the _Central Limit Theorem_) while ensuring linear complexity, resolving the perceived contradiction.

- **Partition Independence (Reviewer cUG3 Q2):** We clarified that under the standard I.I.D. assumption, random partitioning guarantees strict independence, involving **no** theoretical violation.

- **Partitioning Strategy (Reviewer 7jSe Q1,Q5):** We further discussed the influence of partitioning strategies and sizes to address the reviewer's queries.


**4. Experimental Evaluation**

- **Linear Gaussian Settings (Reviewer CASq Q5):** We clarified our simulation setup follows **standard practice**. New experiments in Linear/Gaussian settings show saturated performance (~100% power for all methods), justifying our focus on more complex cases.

- **Real-World Scenarios (Reviewer CASq Q6; cUG3 W3):** We confirmed the **authenticity** and standard usage of our real-world datasets and added small-sample real-world experiments as suggested.

- **Conditioning Set Dimension (Reviewer CASq Q7; cUG3 W3):** We clarified that the challenge of dimension​ $d_Z$ is **orthogonal** to sample size $n$, and pointed to Appendix E.3 where relevant experiments were already presented.

- **Bandwidth (Reviewer nRmx W3):** We justified the use of the standard Median Heuristic across kernel-based methods to ensure a **fair, controlled comparison**, avoiding potential confounders introduced by method-specific optimization.


In summary, we believe all reviewer concerns have been **solidly resolved**. We hope this summary assists you in making a positive assessment.

Thank you again for your contribution to the community.

Best regards, The Authors

---

### Meta-Review · Area_Chair_52hG · 2026-01-06

**Summary:**

The paper proposes Ensemble Conditional Independence Test (E‑CIT), a generic "plug‑and‑play" framework that speeds up any base conditional‑independence test (CIT) by partitioning the full dataset into (K) equal‑size subsets, applying the base CIT on each subset, and then combining the resulting p‑values with a stable‑distribution‑based statistic.

The current work extends existing work in a few important ways: (a) it adds to the literature on ensemble‑based conditional‑independence testing; (b) it provides a new stable‑distribution p‑value combination rule that is distinct from Stouffer’s method and from earlier stable‑distribution‑based tests such as those of Ling & Rho (2022); (c) it achieves linear‑time computational complexity for any base CIT when the subset size is held fixed, a scalability improvement not offered by existing specialized methods like the new FastKCI; (d) may be used with a broad family of CITs.

**Reviewer Concerns:**

**Strengths**
- All reviewers acknowledge that the paper addresses an important practical bottleneck in constraint‑based causal discovery by reducing the computational cost of conditional‑independence tests.
- All find the manuscript well‑written and clearly presented.
- All note that the proposed ensemble framework is compatible with a variety of base CI tests and that the empirical results (speed‑up and competitive power) are convincing.
- The consensus statement explicitly says that "all reviewers acknowledge the practical relevance of reducing CIT computational cost and find the paper well‑written."

**Weaknesses**
- Scope of evaluation: Reviewer CASq criticizes the limited evaluation on linear models, Gaussian noise, and high‑dimensional conditioning sets; nRmx emphasizes the need for a principled method to select the subset size ($n^{(k)}$); cUG3 points out that the experiments focus on synthetic data and standard causal tasks and lack evidence for high‑dimensional or noisy real‑world domains. Reviewer 7jSe does not raise this issue but instead focuses on hyper‑parameter tuning.
- Novelty of the p‑value combination: Reviewer nRmx questions whether the combination rule is truly novel, noting similarities to prior work (Ling & Rho 2022, Liu & Xie 2020). Reviewer 7jSe does not dispute novelty but asks for guidance on choosing the hyper‑parameters ($\alpha$), $(n^{(k)})$, and (K).

**Reviewer Scores:**

Rebuttals:
- CASq’s concern about the lack of linear‑model, Gaussian‑noise and high‑dimensional conditioning experiments: The authors added new experiments with linear causal functions (showing power = 1.0) and with Gaussian‑noise data, reporting that performance is identical to the non‑linear case; they argue this justifies their focus on non‑linear, heavy‑tailed settings. They also clarified that the Sachs flow‑cytometry data are real observational measurements (the “stimulation” refers to biological reagents, not synthetic generation) and included results on the smaller 853‑sample cd3cd28 observational subset.
- nRmx’s request for a principled way to choose the subset size ($n^{(k)}$): The authors supplied a concrete guideline (Appendix F): pick the smallest sample size at which the base CIT yields well‑behaved, approximately uniform p‑values (e.g., ($n^{(k)}\approx 400$) for KCIT). They explain that keeping ($n^{(k)}$) fixed relative to the total ($n$) preserves the linear‑time claim while still providing stable subtest results.
- cUG3’s comment that the evaluation is limited to synthetic data and standard causal tasks, lacking high‑dimensional or noisy real‑world evidence: The authors responded by (i) adding a direct comparison with the Cauchy Combination Test (Appendix E.1), showing comparable Type I error and slightly higher power in heavy‑tailed settings; (ii) emphasizing that empirical Figure 7 demonstrates robustness of performance across a wide range of subset sizes (200–1000), indicating stability even as dimensionality grows; and (iii) noting that the real‑world Sachs flow‑cytometry experiments (both the 1 755‑sample and the 783‑sample subsets) constitute noisy, high‑dimensional observational data, thereby providing evidence beyond purely synthetic scenarios.

I feel the questions about the evaluation were partially answered (more experiments would have been better, of course).

I feel the question about **high‑dimensional conditioning experiments** was not answered adequately. But the others look OK. I feel the reviewers may have raised their scores sligthly.

**A reason to reject the paper**: Another important point is that even in the rebuttal, authors say E‑CIT is the a "generic" ensemble‑based CIT framework, which is a bit of an overclaim... stable distributions are still required, their convergence being slower than Gaussian convergence is an issue, and in real life distributions have parameter drifts. These are not addressed. Not a problem, but the paper **must make its scope clear**, otherwise it stymies future developments that may address these challenges.

---

> ### Public Comment · ~Zhengkang_Guan1 · 2026-02-27
> **Thank you and Camera-Ready Updates**
>
> Dear Area Chair and Reviewers,
>
> We would like to express our sincere gratitude for your constructive feedback and guidance throughout the review process. Your insights have been invaluable in improving the quality of our work.
>
> Following the AC's final recommendations, we have thoroughly revised the camera-ready version to make the scope of the E-CIT framework clearer. Specifically, we have:
>
> 1. Replaced the term "generic" with "general-purpose" throughout the manuscript to prevent any potential misinterpretation or overclaiming.
>
> 2. Restructured and expanded the Discussion section in the main text to explicitly define the applicability, theoretical boundaries, and conditions of the E-CIT framework.
>
> 3. Added dedicated discussions in Appendix F addressing the specific challenges of "Distribution Drifts" and "The Curse of Dimensionality."
>
> Thank you again for your time, effort, and support.
>
> Best regards, Authors

---

### Decision · Program_Chairs · 2026-01-26

Accept (Poster)